# Two-stage electro–mechanical coupling of a K$_V$ channel in voltage-dependent activation

Panpan Hou [1], Po Wei Kang[1], Audrey Deyawe Kongmeneck[2], Nien-Du Yang [1], Yongfeng Liu[1], Jingyi Shi[1], Xianjin Xu[3], Kelli McFarland White[1], Mark A. Zaydman[4], Marina A. Kasimova [2], Guiscard Seebohm[5], Ling Zhong[1], Xiaoqin Zou[3], Mounir Tarek[2]* & Jianmin Cui[1]*

In voltage-gated potassium (K$_V$) channels, the voltage-sensing domain (VSD) undergoes sequential activation from the resting state to the intermediate state and activated state to trigger pore opening via electro–mechanical (E–M) coupling. However, the spatial and temporal details underlying E–M coupling remain elusive. Here, utilizing K$_V$7.1's unique two open states, we report a two-stage E–M coupling mechanism in voltage-dependent gating of K$_V$7.1 as triggered by VSD activations to the intermediate and then activated state. When the S4 segment transitions to the intermediate state, the hand-like C-terminus of the VSD-pore linker (S4-S5L) interacts with the pore in the same subunit. When S4 then proceeds to the fully-activated state, the elbow-like hinge between S4 and S4-S5L engages with the pore of the neighboring subunit to activate conductance. This two-stage hand-and-elbow gating mechanism elucidates distinct tissue-specific modulations, pharmacology, and disease pathogenesis of K$_V$7.1, and likely applies to numerous domain-swapped K$_V$ channels.

[1] Department of Biomedical Engineering, Center for the Investigation of Membrane Excitability Disorders, Cardiac Bioelectricity and Arrhythmia Center, Washington University in St. Louis, St. Louis, MO 63130, USA. [2] Université de Lorraine, CNRS, LPCT, 53000 Nancy, France. [3] Dalton Cardiovascular Research Center, Department of Physics and Astronomy, Department of Biochemistry, Informatics Institute, University of Missouri – Columbia, Columbia, MO 65211, USA. [4] Department of Pathology and Immunology, Washington University School of Medicine, St. Louis, MO 63110, USA. [5] Institute for Genetics of Heart Diseases (IfGH), Department of Cardiovascular Medicine, University Hospital Münster, 48149 Münster, Germany. *email: mounir.tarek@univ-lorraine.fr; jcui@wustl.edu

Voltage gated K+ (K$_V$) channels are homo-tetrameric proteins which sense changes in membrane potential and open the pore to conduct K+ ions through the membrane to regulate cellular excitability. K$_V$ channels contain the voltage-sensor domains (VSDs) formed by the transmembrane segments S1–S4 and the pore domain (PD) formed by the segments S5–S6[1,2]. The VSD activates in a stepwise manner from the resting to the intermediate state before finally arriving at the activated state[3–14], which triggers pore opening via electro–mechanical (E–M) coupling[15–19]. The primary structural determinant underlying the E–M coupling process has classically been assigned to the linking helix between the VSD and the PD (S4-S5 linker or S4-S5L), with the physical movement of the VSD exerting an allosteric tug on key elements of the PD to induce pore opening[15–20]. However, when and where the VSD activation couples to the pore opening still remains elusive. Here, we elucidate a two-stage E–M coupling mechanism in voltage-dependent gating of K$_V$7.1 channels, triggered by the movement of VSD to the intermediate and then to the activated state.

K$_V$7.1, also known as KCNQ1, serves as a key player in regulating cardiac excitability. In the heart, K$_V$7.1 associates with the auxiliary subunit KCNE1 to provide the slow delayed rectifier potassium current ($I_{Ks}$), which is critical for terminating the cardiac action potential[21–24]. Mutation-induced functional defects in K$_V$7.1 are associated with long QT syndrome (LQTS), short QT syndrome, and atrial fibrillation[25,26]. Our previous studies have demonstrated that the E–M coupling plays a key role in K$_V$7.1 channel pathology[5,27]. Deeper understanding of K$_V$7.1 E–M coupling will illuminate mechanistic insights into how inherited mutations impacting K$_V$7.1 lead to disease.

K$_V$7.1 adopts the canonical structural organization of the homo-tetrameric K$_V$ superfamily, and shares significant functional similarity with other K$_V$ members such as the Shaker channel. For example, K$_V$7.1 VSD movements show two distinct steps and conducts ionic current when its VSDs occupy the activated conformation[5,6]. However, uniquely, K$_V$7.1 also conducts ionic current when its VSDs occupy the intermediate conformation[5,6,27]. To distinguish between these distinct K$_V$7.1 conductive states, we refer to the open state associated with the fully activated VSD as the activated-open (AO) state; and the open state associated with the intermediate VSD as the intermediate-open (IO) state. K$_V$7.1 is the only known example for which both the IO and AO states are readily detectable, which presents an opportunity to probe E–M coupling when the VSDs adopt the intermediate vs. the activated conformation.

The traditional paradigm in which VSD movement allosterically opens the pore through the S4-S5L offers a straightforward framework to understand voltage-dependent activation in K$_V$7.1, but several limitations are evident. From a structural standpoint, K$_V$7.1 adopts the common domain-swapped channel architecture of many potassium channels, in which the VSD of each subunit makes substantial contact with the PD of a neighboring subunit[2]. These interactions hint at potential alternative E–M coupling interactions beyond the S4-S5L[17,28]. Previous studies suggest that the two open states (IO and AO) in K$_V$7.1 require distinct E–M coupling mechanisms[5]; however, whether and how the classic E–M coupling interactions contribute to the two open states remains unclear. From a physiological perspective, the classic framework is limited in its ability to explain disease-associated channel mutations. Some LQTS-associated mutations abolish both K$_V$7.1 open states to yield a non-conductive channel[29], while other mutations specifically ablate only one of the two K$_V$7.1 open states[5,6]. Structurally, the hundreds of known K$_V$7.1 disease mutations can be found throughout the channel, including both the VSD and the PD[25]. Yet the mechanistic basis underlying how each mutation leads to channel dysfunction and disease cannot be simply inferred from their location. For example, a mutation within the VSD may affect either VSD stability, VSD activation, and/or E–M coupling to cause channel dysfunction[30,31]. These knowledge gaps in K$_V$7.1 voltage-dependent activation limit our ability to explain the molecular mechanisms by which these mutations disrupt channel function, limiting the power of precision-medicine approaches to address diverse LQTS-associated mutations.

Here, we use the unique properties of K$_V$7.1 and a set of pharmacological and genetic tools[5,6,27] to elucidate a two-stage E–M coupling in voltage-dependent activation. Biophysical measurements and structure-based simulations show that each step of VSD activation evokes a distinctive set of E–M coupling interactions to the pore. When the S4 segment transitions to the intermediate state, the hand-like C-terminus of the S4-S5L interacts with the pore in the same subunit. When S4 then proceeds to the fully activated state, the elbow-like hinge between S4 and S4-S5L also engages with the pore to activate conductance. Because tissue-specific auxiliary subunits of Kv7.1 differentially modulate the IO and AO states to render either open state dominant in different tissues[6], our methods and results provide a foundation to understand physiological role, disease pathogenesis, and specific pharmacological modulation of Kv7.1 in distinct tissue contexts. The two-stage hand-and-elbow gating mechanism likely applies to numerous K$_V$ channels with the domain-swapped architecture.

## Results

**Intermediate VSD activation engages classic E–M coupling.** Previous studies suggested that interactions between the S4-S5L and the cytosolic end of S6 (S6c) are important for the E–M coupling[15,16,18,19]. We tested this mechanism in the K$_V$7.1 channel and found that it is also important for E–M coupling in K$_V$7.1 channels (Fig. 1). To identify these E–M coupling interactions, we carried out voltage-clamp fluorometry (VCF) experiments, a technique in which a fluorophore is covalently attached to the S3-S4 linker of the pseudo-wild type K$_V$7.1 channel (K$_V$7.1-C214A/G219C/C331A, denoted as K$_V$7.1*, Fig. 1a) to monitor VSD movements, while the ionic currents report pore opening[4–7,29]. In K$_V$7.1*, the fluorescence–voltage (F–V) relationship is well-fit by a double Boltzmann function (F$_1$–V and F$_2$–V), which corresponds to VSD activation from resting to intermediate (F$_1$–V) and intermediate to activated (F$_2$–V) states, respectively (Fig. 1a). The two steps VSD activation and two open states gating mechanism of KCNQ1 channels can be described by a simplified kinetic scheme without considering that K$_V$7.1 channels are formed by four subunits with four identical VSDs[5,6,27] (Fig. 1b).

We studied a LQTS-associated mutation V254M[32]. The V254M mutant channel featured strong fluorescence signals associated with two-step VSD movement, with F–V curves exhibiting both F$_1$–V and F$_2$–V components (blue curves; Fig. 1c) comparable to pseudo-WT K$_V$7.1* F–V curves (gray curves; Fig. 1c)[5,6]. However, the observed VSD movements no longer induce pore opening (Fig. 1c), suggesting that E–M coupling interactions in both IO and AO states are severely disrupted. The functional effects of V254M are illustrated in a cartoon in Fig. 1d. We undertook further site-directed mutagenesis in search for more residues with similar phenotypes as V254M. We identified four other mutations, including another LQTS mutation A341V, that altered channel function similarly as V254M (Fig. 1e–g). These five mutants (V254M, H258W, A341V, P343A, and G345A) cluster near the C-terminal ends of S4-S5L and S6c in an arrangement consistent with the classic structural determinants for E–M coupling in other K$_V$ channels[15–19] (Fig. 1h).

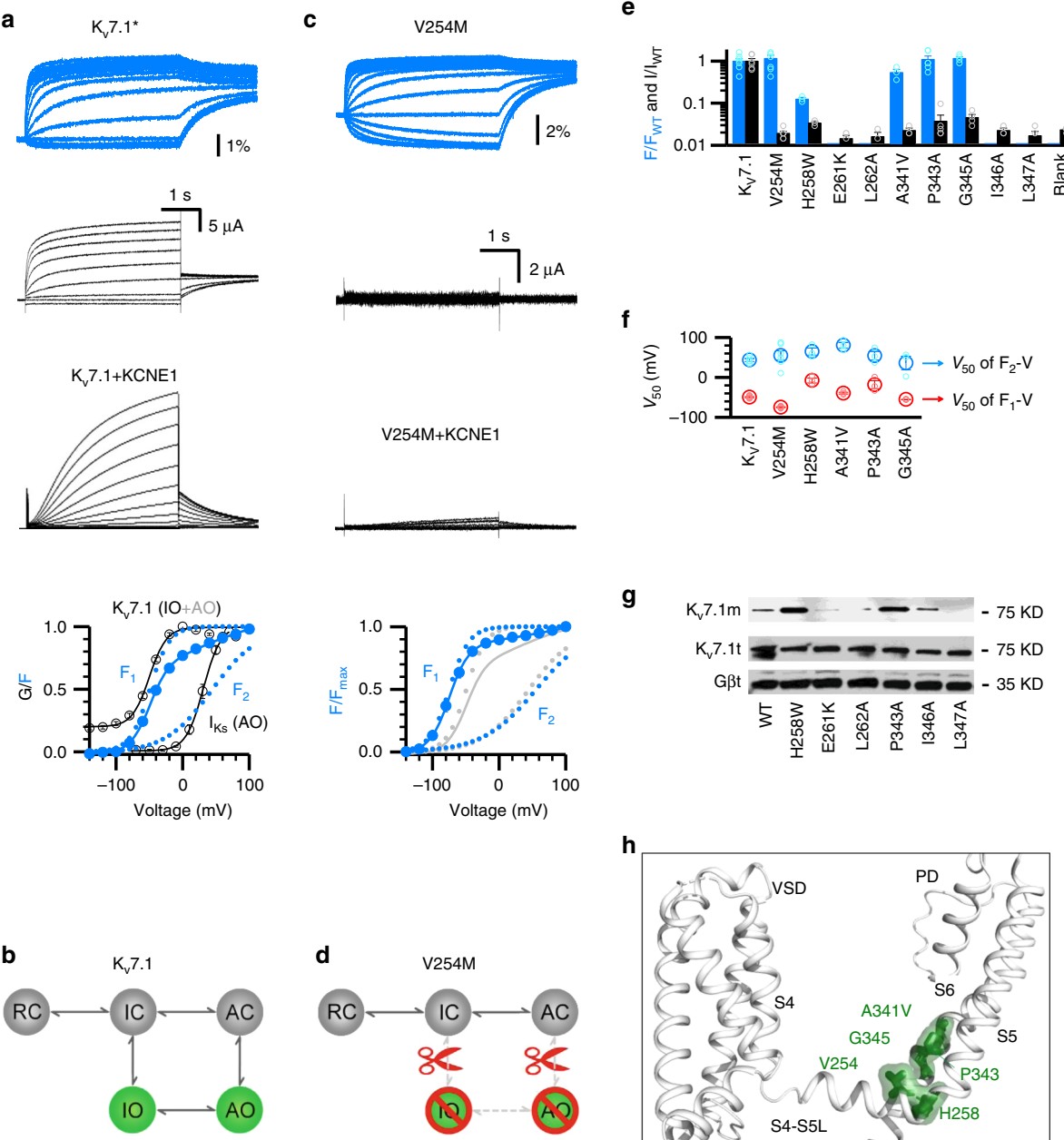

**Fig. 1 Classic interactions are necessary for E–M coupling when the VSD transitions into the intermediate state. a** VCF recordings of pseudo-WT $K_V7.1^*$ ($K_V7.1$-C214A/G219C/C331A). $K_V7.1$+KCNE1 currents are shown at the same scale. The F–V relationships (blue circles) are fitted with a double Boltzmann function. $F_1$–V represents the VSD transition from resting to intermediate state; $F_2$–V represents the VSD transition from the intermediate to activated state. G–V represents channel opening with VSD transition at both the intermediate (IO) and the activated (AO) states. **b** Cartoon scheme illustrating the gating mechanism of two-step VSD movements and distinct two open states. **c** VCF recordings of V254M in $K_V7.1^*$. V254M+KCNE1 currents are shown at the same scales. The F–V relationships of $K_V7.1^*$ and V254M are shown in gray and blue, respectively. **d** Cartoon scheme illustrating that V254M disrupts E–M couplings for both IO and AO. **e** Summary of data for WT and mutant $K_V7.1$. VSD activation (blue, percentage change in fluorescence) and pore opening (black, current amplitude) are normalized to the WT. n ≥ 3. Blank: cells not injected with channel mRNA. Data points are shown in small open circles. **f** $V_{50}$ values for the $F_1$–V and $F_2$–V. n ≥ 3. Data points are shown in small open circles. **g** Western blot results showing the membrane (top) and total (middle) expression of some mutants that eliminated both fluorescence and ionic currents. Gβ (bottom) from total protein is shown as negative control. WT $K_V7.1$, H258W, and P343A are shown as positive controls. **h** Mapping the key residues V254, H258, A341, P343, and G345 (green) onto the S4-S5L/S6c interface in the $K_V7.1$ cryoEM structure (PDB: 5VMS)[2]. All averaged data are shown in mean±SEM. Source data are provided as a Source Data file.

Some additional mutations in this region (E261K, L262A, I346A, and L347A) yielded no currents or fluorescence signal due to reduced channel expression (Fig. 1g), precluding functional analysis for E–M coupling.

These results suggest that in $K_V7.1$ the classic E–M coupling interactions are also necessary for pore opening. Critically,

mutational disruption of the classic E–M coupling eliminates $K_V7.1$ opening at both the IO and AO states (Fig. 1c–h), suggesting that the classic E–M coupling interactions are already engaged upon VSD transition into the intermediate state and maintained at the activated state. Moreover, these interactions are necessary for pore opening at both IO and AO states.

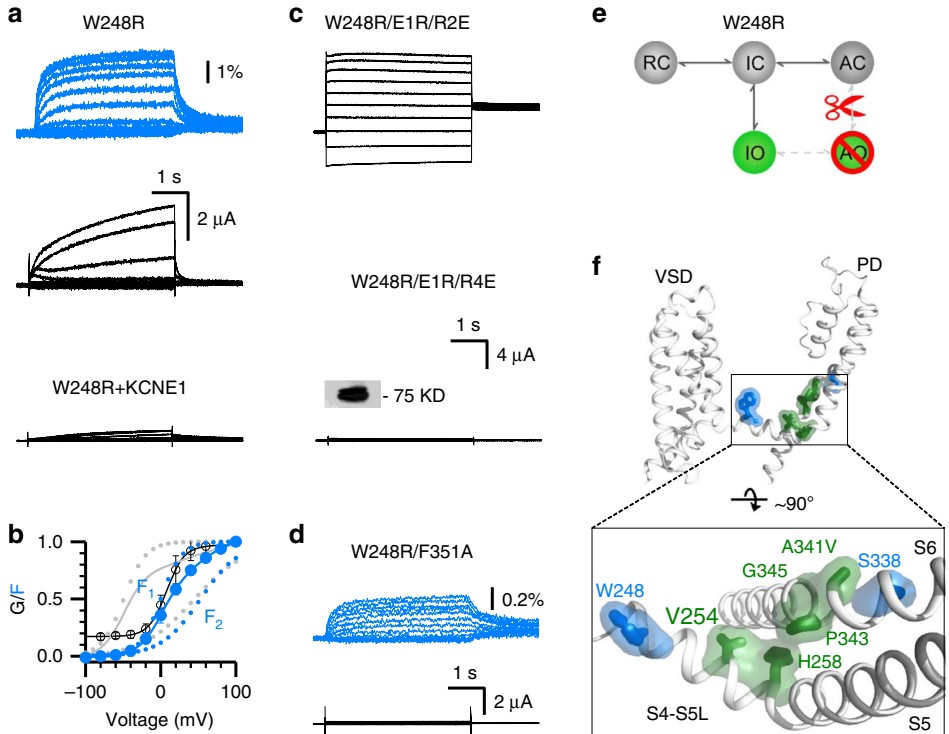

**Fig. 2 LQTS mutation W248R specifically eliminates the E–M coupling when the VSDs adopt the activated conformation. a**, **b** VCF recordings of W248R in $K_V7.1^*$. W248R+KCNE1 currents are shown at the same scale. The F–V relationships of $K_V7.1$ and W248R are shown in gray and blue, respectively. $n \geq 3$. All averaged data are shown in mean±SEM. **c** Representative currents of W248R/E1R/R2E and W248R/E1R/R4E activated from −120 mV to 60 mV with 20 mV increments. Same scale for both currents. The inset shows western blot data for membrane expression of W248R/E1R/R4E. **d** VCF recordings of W248R/F351A. **e** Cartoon scheme illustrating that W248R specifically disrupts the AO E–M coupling. **f** Mapping the key residues at the S4-S5L/S6c interface (green, V254, H258, A341, P343, and G345) in Fig. 1, and W248 and S338 (blue) onto the $K_V7.1$ cryoEM structure (PDB: 5VMS)[2]. Source data are provided as a Source Data file.

Then, are these classic interactions also sufficient for $K_V7.1$ pore opening?

**Activated VSD activation engages AO specific E–M coupling.** In studying another LQTS-associated mutation W248R[33], we found that the mutation specifically disrupted E–M coupling when the VSDs adopt the activated state, but left E–M coupling intact when the VSDs occupy the intermediate state, i.e., W248R selectively disables AO state E–M coupling, resulting in a channel that is conductive only in the IO state.

To determine whether a mutation specifically ablates the AO state, we utilized two extensively validated experimental strategies[5,6,9,27,34,35]. The first tests the effect of the mutations when the VSDs are strongly biased to occupy the intermediate or in the activated states. To this end, the mutation is co-mutated on background intermediate VSD-locked (E160R/R231E, E1R/R2E) and activated VSD-locked (E160R/R237E, E1R/R4E) mutant channels[5,6,9,27,34,35]. A mutation that specifically ablates the AO state should exhibit little ionic current when combined with the activated-VSD-locked (E1R/R4E) channel. On the other hand, when the same mutation is combined with the intermediate-VSD-locked (E1R/R2E) channel, robust IO state ionic current would be expected. The second strategy assays the effect of the mutation upon specific ablation of the IO state. This is achieved by two methods: KCNE1 co-expression and co-mutation with F351A. Both maneuvers suppress IO state conductance in $K_V7.1$[5,6]. A mutation that specifically abolishes the AO state should exhibit no ionic current when co-expressed with KCNE1 or when co-mutated with F351A, as both IO and AO states are

eliminated. If a mutation demonstrates consistent read-out across all four distinct tests (KCNE1 co-expression; co-mutation with E1R/R4E, E1R/R2E, and F351A), then the effect of the mutation will be attributed to selective disruption of the AO state.

W248R exhibited robust ionic current and VSD movement (Fig. 2a, b). When we applied the described four experiments to detect specific AO state disruption, a consistent picture emerges: W248R yields robust ionic current upon co-mutation with E1R/R2E (intermediate-VSD-locked), but co-mutation with E1R/R4E (activated-VSD-locked) results in little ionic current despite robust membrane expression (Fig. 2c). On the other hand, KCNE1 co-expression and F351A co-mutation (IO eliminated) with W248R strongly suppresses ionic current, despite robust fluorescence observed, indicating intact surface membrane expression and VSD activation (Fig. 2a, d). These results demonstrate that W248R specifically eliminates the AO state without eliminating VSD activation. VCF measurements indicate that the VSD of W248R still activates in two resolvable steps, with both $F_1$–V and $F_2$–V components (Fig. 2b), demonstrating that W248R does not affect the VSD transition to the activated state. These results are consistent with the mechanism that W248R specifically disrupts E–M coupling of the AO state, as illustrated in Fig. 2e.

The functional effects of W248R on $K_V7.1$ gating are similar to another LQTS-associated mutation S338F[5]. Structurally, W248 and S338 are located in the N-terminus of S4-S5L and the middle of S6, respectively, both of which are located generally outside of the region of the classical E–M coupling (Fig. 2f). Taken together, these results suggest that the classic E–M coupling interactions, while necessary, are not sufficient for pore opening when the VSD is at the activated state. Another set of E–M coupling interactions

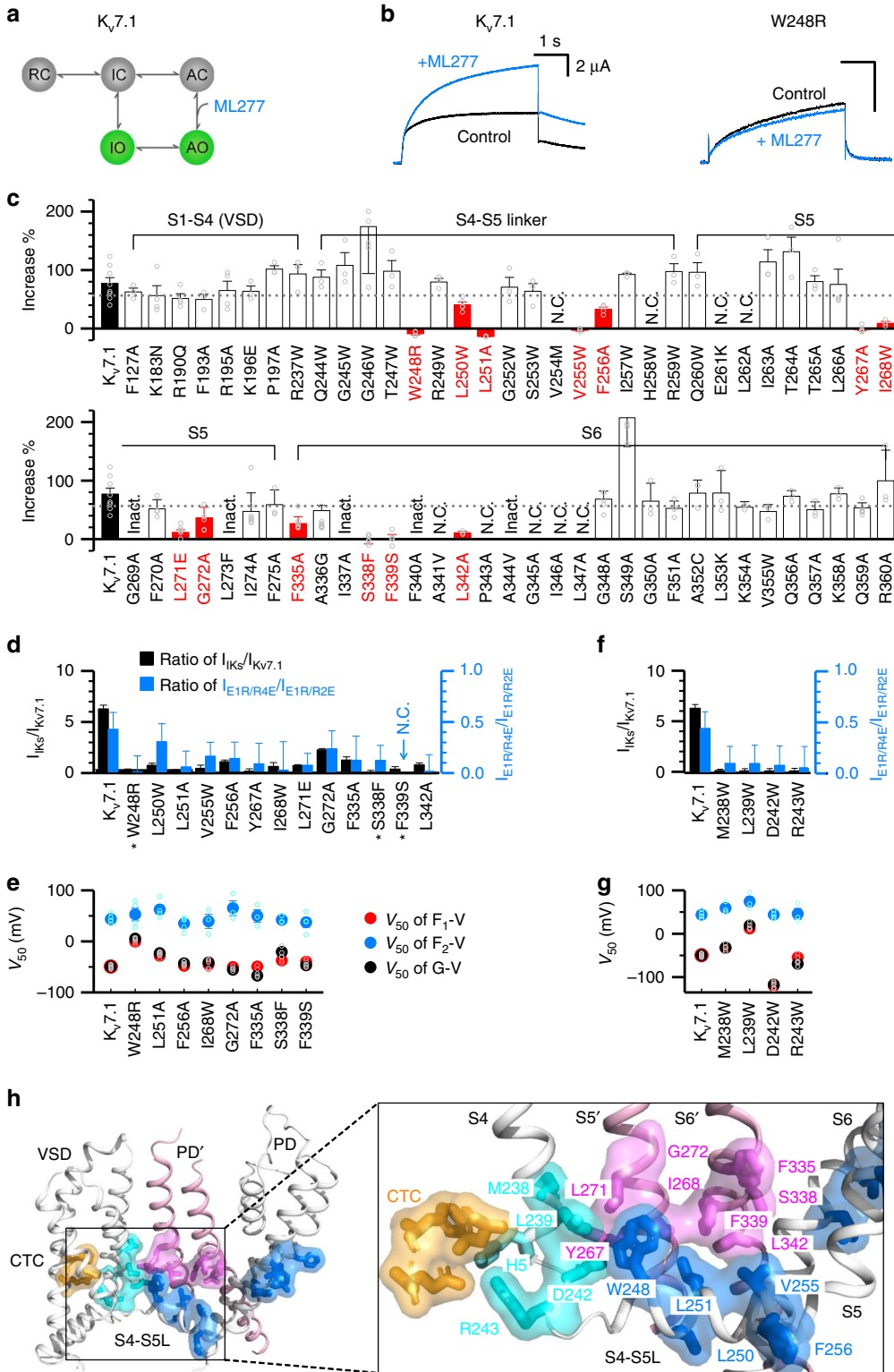

are required for conduction when VSDs move from the intermediate to the activated state. Mutations such as S338F and W248R that disrupt these interactions result in selective loss of AO-current and are associated with arrhythmias. Encouraged by this finding, we set out to experimentally map this second set of specific E–M coupling interactions.

To identify residues involved in this second set of AO state E–M coupling interactions, we developed a pharmacological assay by utilizing a small molecule $K_V7.1$ activator ML277[36,37], which increases $K_V7.1$ current by specifically potentiating the AO state E–M coupling[27] (Fig. 3a, b). This unique mechanism provides a straightforward assay: mutations that disrupt AO state E–M coupling (e.g. W248R) or ML277 binding (Supplementary Fig. 1) would result in loss of the ML277 potentiation of $K_V7.1$ currents[27] (Fig. 3b). Utilizing this strategy, we combined ML277 with scanning mutagenesis across the channel (Fig. 3c). For sites of known disease mutations, it was the disease mutant forms that were examined, while for all other sites the mutations

**Fig. 3 Key residues involved in the E–M coupling when the VSD adopts the activated conformation. a** A cartoon scheme to illustrate that ML277 specifically enhances AO state E–M coupling[27]. **b** Currents of WT $K_V$7.1 and W248R before and after adding 1 μM ML277. Voltage: +40 mV then returned to −40 mV. **c** 1 μM ML277-induced current increase on $K_V$7.1 WT and with mutations in the VSD, S4-S5L, S5, and S6. The residues were mutated to alanine (A), tryptophan (W), or to known LQTS mutations. The dotted line shows 2X the standard error below the average current increase of WT $K_V$7.1. Mutations that show ML277-induced current increases (mean + SEM) below the dotted line are labeled as red. N.C. mutations show little or no current. Inact. mutations show obvious c-type like inactivation[5]. Data points are shown in small open circles. **d** Current ratios of $I_{IKs}/I_{Kv7.1}$ (black) and $I_{E1R/R4E}/I_{E1R/R2E}$ (blue) for WT and mutant $K_V$7.1. Star indicates LQTS-associated mutations. **e** $V_{50}$ values of G–V, $F_1$–V, and $F_2$–V relations for WT and mutant $K_V$7.1 channels. Data points are shown in small open circles. **f** Current ratios of $I_{IKs}/I_{Kv7.1}$ (black) and $I_{E1R/R4E}/I_{E1R/R2E}$ (blue) for WT and the S4c mutant channels. **g** $V_{50}$ values of G–V, $F_1$–V, and $F_2$–V relations for WT and S4c mutant channels. Data points are shown in small open circles. **h** Mapping the residues key to E–M coupling for AO state onto the $K_V$7.1 cryoEM structure[2]. Only part of two adjacent subunits are shown. Yellow marks the residues in the charge transfer center (CTC) F167 (F0), E170, and D202. S1–S3 are transparent in the inset for clarity. Cyan: the four residues at the S4c, the fifth gating charge H5 (without showing the surface) is in the CTC when the S4c adopts the activated conformation[2]; blue: the thirteen residues from the ML277-screening. Pink shows eight residues from the neighboring subunit S5' and S6' from the ML277-screening. All averaged data are shown in mean ±SEM. Source data are provided as a Source Data file.

were to alanine or tryptophan. This strategy revealed 13 mutations, including W248R and S338F, that eliminated or reduced the ML277 potentiation effect (red bars; Fig. 3c). Of the thirteen residues, five are in the S4-S5L (W248, L250, L251, V255, and F256), four are in the S5 helix (Y267, I268, L271, and G272), and four are in the S6 helix (F335, S338, F339, and L342).

To confirm that these 13 mutations indeed selectively disrupted AO state E–M coupling, we performed the same functional assay afforded to W248R (Fig. 2) on the 11 mutations. For all these mutations, co-mutation with intermediate-VSD-locked (E1R/R2E) channels demonstrated robust current; while co-mutation with activated-VSD-locked (E1R/R4E) channels yielded only low ionic current (Fig. 3d, Supplementary Fig. 2). Suppressing the IO state with KCNE1 co-expression and F351A co-mutation also resulted in reduced current amplitude in all these mutants, although VCF experiments (S338F+KCNE1 and L251A+KCNE1) indicated membrane expression and VSD function (Fig. 3d, Supplementary Fig. 2). Altogether, these functional results provided a consistent read-out across four independent experiments, suggesting that these thirteen mutations indeed selectively disrupt the AO state. For all the mutant channels for which we were able to measure VCF signals, the results exhibit a two-step VSD activation with both F1–V and F2–V components (red and blue circles; Fig. 3e, Supplementary Fig. 2). Moreover, the voltage dependence of pore opening (G–V relation) tightly follows the F1–V relationship, consistent with the hypothesis that these mutant channels only conduct at the IO state (black and red circles; Fig. 3e). The VCF data indicate that these mutants do not prevent VSD transition into the activated states, suggesting that these residues are critical for the AO state E–M coupling when the VSD adopts the activated conformation.

Interestingly, we previously found four mutations (M238W, L239W, D242W, and R243W) at the C-terminal end of the S4 segment (S4c) that also showed robust currents, but the currents were severely suppressed by the co-expression of KCNE1[34]. Further functional studies show that these S4c mutations produce similar phenotypes as the residues identified from the ML277 screen[34] (Fig. 3f, g, Supplementary Fig. 3), suggesting that the S4c segment is also directly involved in E–M coupling for the AO state.

Mapping all these 17 residues onto the $K_V$7.1 cryo-EM structure[2] reveals that they form two clusters in each subunit, which are spatially distinct from the classic E–M coupling interface (Figs. 1h, 3h). One cluster is located in S4c (cyan) and S4-S5L (blue), and the other is in S5 and S6 (blue; Fig. 3h). Within the VSD, key coupling residues at the S4c (cyan) are located adjacent to the hydrophobic plug or charge transfer center[38,39] (CTC, orange; Fig. 3h). The VSD in the cryo-EM structure adopts the activated state with the fifth gating charge

H240 (H5, cyan stick without surface) engaged in the CTC[2,6,9], suggesting that the movement of S4 to the activated state engages the S4c and S4-S5L into the CTC thereby initiating these AO-state specific E–M coupling interactions. The two clusters of sites are not in proximity within a single subunit, but instead form extensive contacts with clusters from the neighboring subunit, with the cluster in S4c and S4-S5L directly facing the cluster in S5 and S6 from an adjacent subunit (pink; Fig. 3h). This spatial arrangement maps network of residues stretching from S4 all the way to the pore, forming an E–M coupling pathway (Fig. 3h). Interactions among these residues are exclusively for the AO state E–M coupling when the VSD adopts the activated conformation.

To experimentally confirm that the above E–M coupling residues indeed interact during $K_V$7.1 voltage-dependent activation, we utilized double mutant cycle (DMC) technique, a method extensively applied to residue interactions in proteins[40]. DMC has previously been used to identify interacting residue pairs in ion channels[41–43], by measuring the net free energy change (ΔΔG) associated with activation that occurs when the interaction between two residues is disrupted. Specifically, DMC measures the energy changes of the WT channel ($G_0$), two single mutants ($G_1$ and $G_2$), and the corresponding double mutant ($G$). The two residues are identified as interacting if the energy changes resulting from the two single mutants do not linearly sum to that observed in the double mutant ($|\Delta\Delta G| = |\Delta G_1 + \Delta G_2 - \Delta G| \geq 1$ kcal/mol). On the other hand, the two residues are classified as not interacting if $|\Delta\Delta G| < 1$ kcal/mol[43].

We applied the DMC technique for the residue pair L251 (S4-S5L) and I268 (S5), which appear to be spatially close at the interface between two neighboring subunits. The activation energies of voltage-dependent activation of the WT ($G_0 = 1.7$ kcal/mol), L251A ($G_1 = 0.1$ kcal/mol), I268A ($G_2 = 1.2$ kcal/mol), and L251A/I268A ($G = -2.2$ kcal/mol) channels were non-additive (Fig. 4a, b), with a net energy change $\Delta\Delta G = 1.8$ kcal/mol (>1 kcal/mol). The result suggests a direct interaction between L251 and I268. Besides DMC, the gating kinetics also supports this interaction: I268A exhibits an obvious inactivation phenotype, suggesting a substantial change on the gating process. However, the double mutant L251A/I268A abolishes this inactivation phenotype from I268A, which indicates that the inactivation phenotype of I268A is not additive in L251A/I268A, and that L251A rescues the gating change seen for I268A (Fig. 4a). DMC results also support the importance of S4-S5L/pore interactions (W248/I268 and L251/F339), and S4c/S5 interactions (M238/L271 and L239/L271) (Fig. 4c, d, Supplementary Fig. 4).

**Two-stage E–M coupling mechanism.** Our results so far indicate that $K_V$7.1 features two spatially distinct sets of E–M coupling

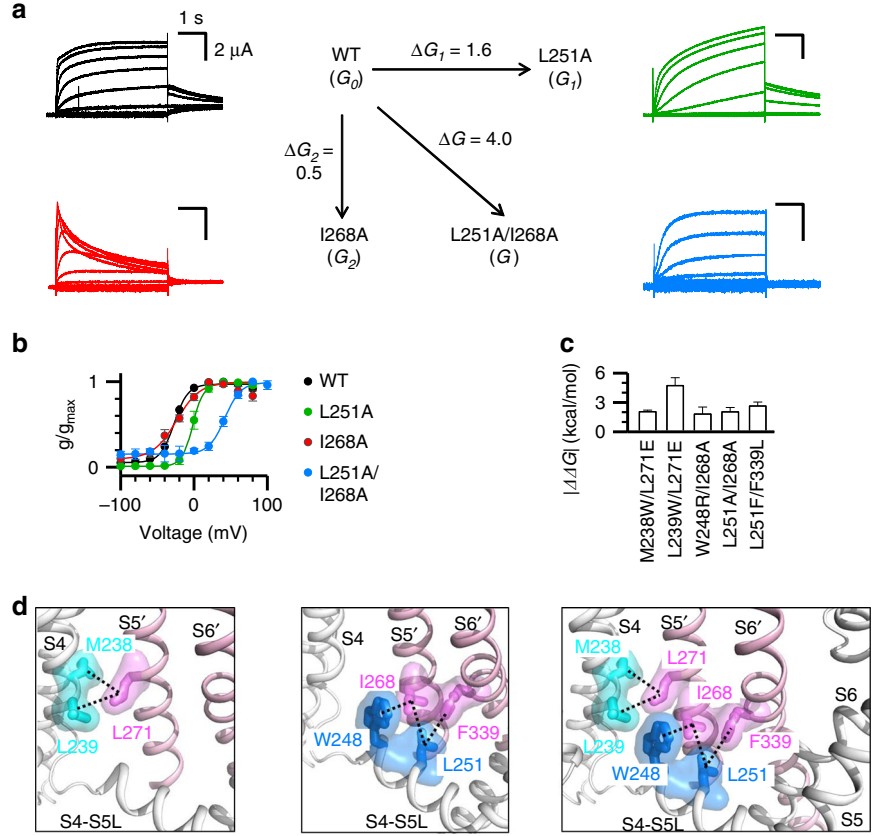

**Fig. 4 E-M coupling interactions when the VSD adopts the activated conformation. a, b** Double mutant cycle analysis of the interaction between L251 and I268. $\Delta\Delta G = \Delta G - (\Delta G_1 + \Delta G_2) = 1.8$ kcal/mol, which indicates interaction between the two residues. $n \geq 3$. All averaged data are shown in mean ± SEM. **c** Double mutant cycle analysis of interactions between W248 and I268, L251 and I268, L251 and F339, M238 and L271, and L239 and L271 ($\Delta\Delta G = 1.4$–4.4 kcal/mol). $n \geq 3$. **d** Mapping the five pairs of interacting residues onto the $K_V$7.1 cryoEM structure. Color codes are the same as in Fig. 3h. Source data are provided as a Source Data file.

interactions: (1) the classic set of interactions at the S4-S5L/S6c interface (Fig. 1), which are engaged when the VSD moves to the intermediate state and maintained at the activated state, and (2) the set of interactions at the S4c/S5 and the S4-S5L/pore interfaces (Figs. 2–4) which are specific to the VSD adopting the activated state. We next undertook molecular dynamic (MD) simulations to correlate VSD activation motion to these two sets of E–M coupling interactions. We performed MD simulation on four states of $K_V$7.1 with different combinations of VSD-PD states: resting-closed (RC), activated-closed (AC), IO, and AO (see Methods, Supplementary Fig. 5). In order to quantify possible interactions between residue pairs in each model, we measured the frequency at which the distance between sidechains remain below the threshold values (see Methods) in all four subunits throughout the MD trajectories. Consistent with experiments, the pairs of possible interactions indicated in the MD trajectory include the residues found to be critical for E–M coupling (Figs. 1, 4, 5a–d). MD simulations found that the interactions among classic E–M coupling residues are maintained in all states (Fig. 5a–b, Supplementary Fig. 6), while interactions among the AO-state E–M coupling residues fall into three groups of state-dependent interactions. The first group includes interactions only present in the AO state model, where they may specifically stabilize AO state E–M coupling (Fig. 5c–d; top). The second group includes interactions only absent in the AO model, but present in other models (Fig. 5c–d; bottom). These interactions may need to be broken to enable AO state E–M coupling. The third group includes interaction L251/I268 which is present in both IO and AO states (Fig. 4a, b and Fig. 5c, d). Taken

together, these MD simulation data are consistent with the experimentally identified interactions required for $K_V$7.1 pore conduction. They indicate that the classic interactions are important in the E–M coupling when VSD is at both the intermediate and activated states, while the AO state E–M coupling interactions are engaged upon VSD transition into the activated state.

In sum, our findings in this study reveal a two-stage E–M coupling process and lead to a hand-and-elbow gating mechanism uncovering when and where E–M coupling interactions engage during voltage-dependent activation of $K_V$7.1 channels (Fig. 5e). In this model, the S4 helix and S4-S5L resemble a bent arm. The S4 (upper arm) moves in two resolvable steps, first to the intermediate state and then to the activated state. The sequential movements engage two stages of E–M coupling interactions through distinct mechanisms. (1) The motion of the S4 (upper arm) promotes channel opening through the S4-S5L (hand) grip of the S6c of the same subunit at the first conductive IO state. (2) Subsequently, VSD transitions to the fully activated state, which engages the S4/S4-S5L joint (elbow) in direct interactions with the S5 and S6 of a neighboring subunit to nudge the pore to adopt the second conductive AO state.

To further test if the two-stage hand-and-elbow E–M coupling mechanism is broadly conserved among $K_V$ channels, we performed a statistical coupling analysis (SCA)[44,45]. SCA analyzes covariation in channel protein sequences and identifies groups of co-evolving residues termed protein sectors that correspond to networks of amino acid interactions critical for channel function[46] (Fig. 6). Since $K_V$7.1 channel adopts the domain-

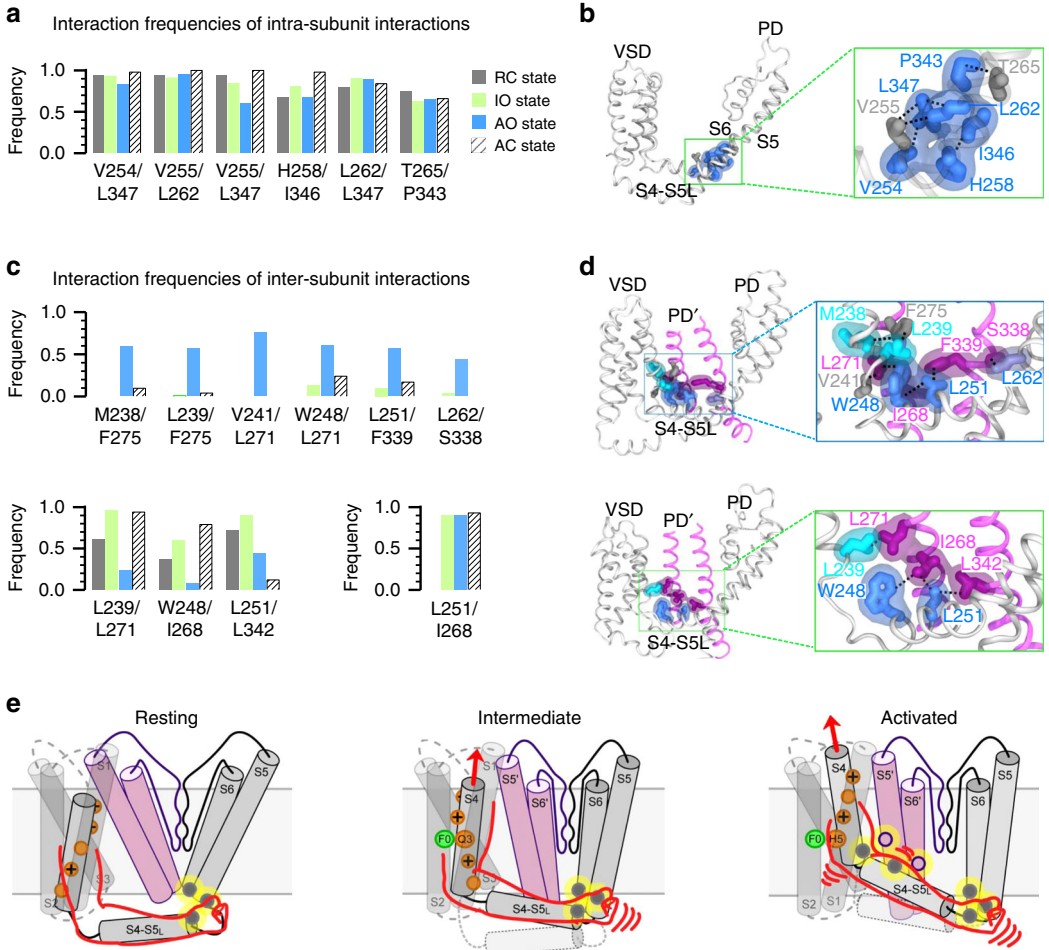

**Fig. 5 MD simulation correlates temporal sequence of VSD movement to engagement of the two-stage E–M coupling interactions. a** Interaction frequencies of $K_V7.1$ residue pairs as observed during the MD trajectories for the $K_V7.1$ RC, IO, AO, and AC models. Frequency threshold of 0.4 or higher is defined as indicative of interaction between two residues (see Methods). **b** Representations of the IO state model of a $K_V7.1$ subunit. Inset: enlarged structures of the boxed area. Blue, residues identified as important for E–M coupling in experiments (Fig. 1c–g); gray: V255 and T265, which were identified in MD simulations, but for which mutations still show functional currents (Fig. 3c). Residue V255 was found to be important for the AO state E–M coupling (Fig. 3). Each pair of residues is represented by a black dashed line connecting their respective sidechains. **c** Interaction frequencies of AO model-specific (top), IO model-specific (bottom left), and in both IO and AO models (bottom right) residue pairs during the MD trajectories. **d** Representations of the AO (top) and IO states (bottom) models of $K_V7.1$. An entire subunit (in gray) as well as S5' and S6' segments of an adjacent subunit (in purple) are shown with insets of enlarged structures of the boxed area. Residues important for E–M coupling in experiments (Fig. 3) and involved in residue pair interactions in MD simulations are colored similarly as in Fig. 3h: cyan, residues in S4c; blue: S4-S5L, and purple: S5 and S6 from the adjacent subunit. Residues V241 and F275, which were identified in MD simulations as possibly involved in sidechain interactions but for which mutations did not alter AO states, are colored in dark grey. Each pair of residues is represented by a black dashed line connecting their respective sidechains. **e** Cartoon schemes illustrating the two-stage hand-and-elbow mechanism of $K_V7.1$ voltage-dependent activation. Only two neighboring subunits (gray and pink) are shown for clarity. Gating charges on the S4 segment are shown in orange, the hydrophobic plug F0 in the gating charge transfer center is shown in green, and the two sets of interactions are shown as yellow circles. Source data are provided as a Source Data file.

swapped architecture[2], we applied SCA to 1,421 domain-swapped $K_V$ sequences (Fig. 6a–b) and identified two protein sectors utilizing SCA methodologies[45] (Fig. 6b–c, see Methods, Supplementary Fig. 7). Sector 1 maps to two physically disconnected inward-facing clusters within the VSD and the PD (red; Fig. 6d). Notably, sector 1 demonstrates a lack of residues within the S4-S5L and a paucity of VSD-PD interactions (Fig. 6d). We interpret sector 1 as mainly including residues that are critical for maintaining the independent stabilities and functions of the VSD and PD domains. Sector 2 maps a network of residues that flows from the VSD to the PD through the S4-S5L reminiscent of classic E–M coupling interactions (blue; Fig. 6e). Moreover, when the same sector is mapped onto the neighboring $K_V7.1$ subunit, numerous inter-subunit interactions between S4,

S4-S5L, and the neighboring S5 and S6 helices can be observed (blue and pink; Fig. 6e). As mentioned earlier, previous studies in Shaker $K^+$ channels identified several pairs of residues that are involved in inter-subunit interactions and contribute to E–M coupling[17,28]. Our experimental and SCA results indicate that the inter-subunit interactions are part of a chain of interactions including residues in S4c, S4-S5L, S5 and S6, and these interactions are specifically important for the AO state E–M coupling. We thus interpret sector 2 as an E–M coupling sector that contains residues involved in both stages of $K_V$ E–M coupling. The SCA results suggest that the $K_V7.1$ E–M coupling interactions in this study may be functionally intact in all domain-swapped $K_V$ channels, and the two-stage E–M coupling process may be broadly conserved.

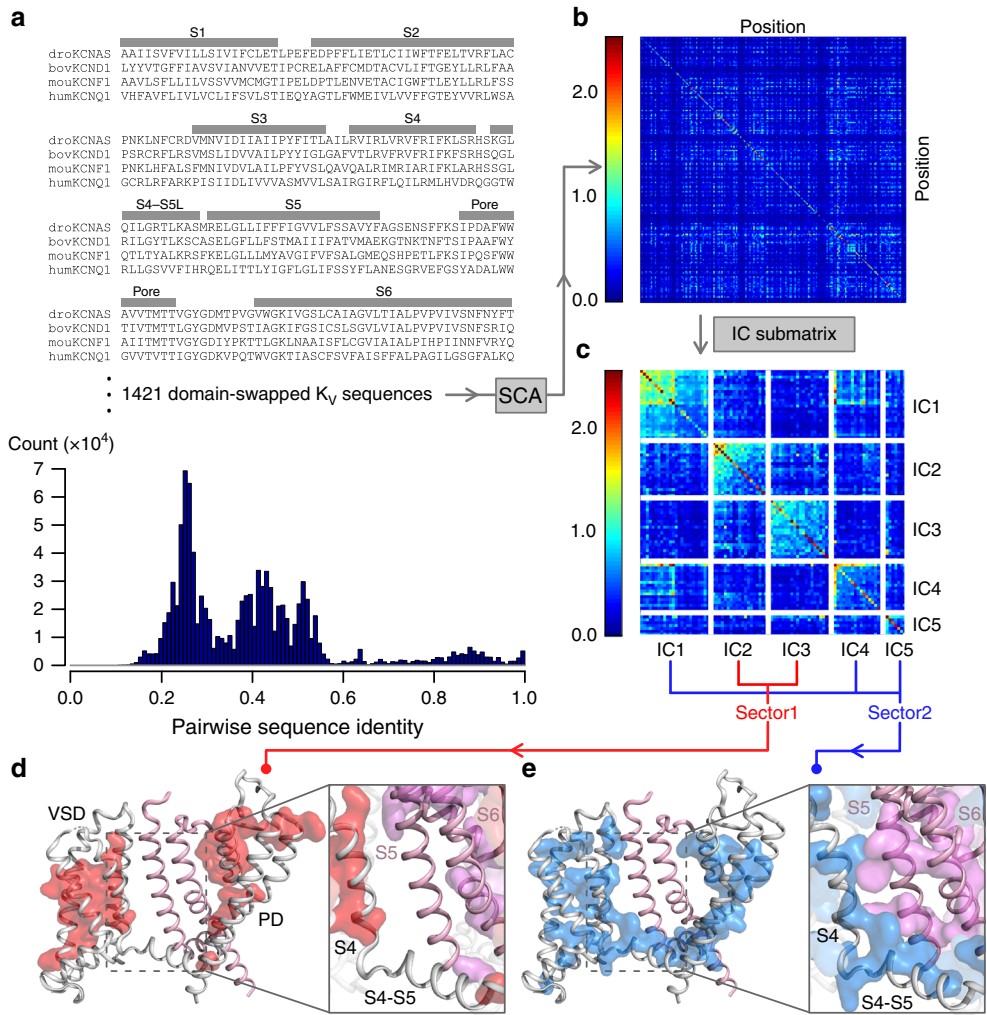

**Fig. 6 Statistical coupling analysis suggest conservation of the two-stage E–M coupling across domain-swapped K$_V$ superfamily. a** Sample multiple sequence alignment (MSA) containing domain-swapped K$_V$ channels (K$_V$1-K$_V$9) used as input for statistical coupling analysis (SCA). The bottom histogram shows counts for pairwise sequence identity between all pairs of residues within the MSA. MSA uses the nomenclature established by the HUGO Gene Nomenclature Committee (HGNC) for voltage-gated potassium channels[80]. **b** SCA covariance matrix calculated by SCA (see Methods). *X* and *Y* axes indicate amino acid positions, while colors indicate the degree of covariance, with blue and red corresponding to lowest and highest degrees of covariation. **c** The independent component (IC) submatrix visualizes a subset of the SCA matrix (panel b) in which the residues are the highest covariant. Each IC represents a group of co-varying/co-evolving amino acids as calculated by SCA (see Methods). Axes are amino acid position numbers. Diagonal boxes indicate covariation within each IC, off-diagonal boxes indicate cross talk between ICs. Sector 1 and sector 2 were defined by combining the IC as shown (also see Methods and Supplementary Fig. 7). **d–e** Mapping sectors 1 and 2, as calculated by SCA, on the K$_V$7.1 cryoEM structure. The main subunit is colored gray, the neighboring subunit is colored pink. Sectors 1 (**d**) and 2 (**e**) on the main subunit are colored red and blue. Insets show S4, S4-S5L of the main subunit and S5, S6 of the neighboring subunit. The respective sectors on the neighboring subunit are colored magenta in the inset. All other elements in the insets are transparent for the clarity. Source data for SCA are provided as a Source Data file.

## Discussion

Elucidating the fundamental mechanism of voltage-dependent gating of K$_V$ channels remains an important goal of basic biomedical research. After decades of structural and functional studies, processes such as VSD activation are now relatively well-understood. However, E–M coupling remains poorly understood, especially in regard to how each of the individual VSD transitions contributes to opening the pore. This is because E–M coupling is dictated by the energetics and dynamics of protein-protein interactions, which cannot be determined directly with structural studies and lack of exclusive approaches to investigate with functional studies. In this study, extensive mutagenesis, pharmacology, voltage clamp fluorometry experiments, and MD simulations provide the two-stage hand-and-elbow mechanism for how the stepwise VSD activation is coupled to the pore and its

two distinct conductive states in K$_V$7.1 channels (Figs. 1–5). In this mechanism, the classic E–M coupling interactions at S4-S5L/S6c promotes channel opening upon VSD movement into the intermediate state, while another set of VSD/pore and S4-S5L/pore interactions engage only upon transition to the fully activated state. This work relied on unique alternative open states (IO and AO) in K$_V$7.1, which present unambiguous current readouts when the VSDs adopt the intermediate vs. activated state conformations. This unique trait also enables functional detection of the E–M coupling interactions explicitly associated with the intermediate and activated VSD states, a feat difficult to achieve in other channels, which only conduct when the VSD occupies the activated state (e.g. Shaker channels)[38,39].

SCA performed in this study suggests that the hand-and-elbow gating model may extend to other domain-swapped K$_V$ channels.

This result is consistent with recent discoveries in Shaker $K^+$ channels that some non-canonical, inter-subunit interactions besides the classic E–M interactions also contribute to E–M coupling[17,28,47,48]. The residues important for the non-canonical interactions identified in these recent studies are among the network of residues revealed in our SCA as being important for E–M coupling (Fig. 6). Although we defined sector 2 (Fig. 6e) in our SCA as an E–M coupling sector, we cannot exclude the possibility that sector 2 also includes residues important for VSD activation or gate opening. However, sector 2's spatial pattern includes abundant residues within the S4-S5 linker which strongly suggests sector 2 contains significant number of E–M coupling residues. SCA was applied to the $K_V$ channel super-family in prior studies; however, the input sequence alignment consisted of both domain-swapped and non-domain-swapped $K_V$ channels[46]. These two architecturally-distinct ion channel super families likely feature distinct E–M coupling mechanisms. For example, the VSD of non-domain-swapped channels do not form inter-subunit contact with the PD of their neighboring subunit. The inter-subunit VSD-Pore E–M coupling mechanism discovered in this study is thus incompatible with the non-domain-swapped $K_V$ architecture. In this light, including both channel architecture types in a single SCA may confound results for protein sector related to E–M coupling.

$K_V7.1$ and KCNE1 complexes form the $I_{Ks}$ channels important in controlling cardiac action potential duration[21–24]. More than 300 mutations of $K_V7.1$ are associated with LQTS, but so far only for a portion of these mutations the mechanism of altering channel function is understood. Our studies of the LQTS mutations that disrupt E–M coupling led to the discovery of two distinct sets of interactions that mediate E–M couplings in $K_V7.1$ channels (Figs. 1–3). Among these mutations, two (V254M, A341V) disrupt the classic E–M interactions, resulting in the total loss of $I_{Ks}$ due to the elimination of both IO and AO states (Fig. 1); while the other three (W248R, S338F, and F339S) disrupt the E–M interactions specific for the AO state, which also result in the total loss of $I_{Ks}$ (Figs. 2, 3) as the $I_{Ks}$ channel only conducts AO-state current[5,6].

Tissue-specific auxiliary KCNE subunits differentially modulate the $K_V7.1$ IO and AO states[6]. In the heart, KCNE1 modulates $K_V7.1$ to only conduct at the AO state[6]. Drugs such as ML277 that specifically enhance the AO state[27] (Fig. 3) may be candidates for specific anti-arrhythmic therapy. Our docking results show that ML277 binds at the S4-S5L/pore interface where the residues interacting with ML277 are critical for the AO state E–M coupling (Fig. 3, Supplementary Fig. 1), serving as an excellent target for developing drugs with AO state specificity. On the other hand, patients who harbor one or more of LQTS-associated mutations that specifically disrupt AO state E–M coupling (Fig. 3d) are likely refractory to treatment with drugs that affect $K_V7.1$ similarly as ML277 (Figs. 2, 3). The two-stage hand-and-elbow model thus furnishes a framework to conceptualize $K_V7.1$ auxiliary subunit regulation, tissue-specific physiology, disease pathogenesis, and state-dependent pharmacological modulation.

## Methods

**Constructs and mutagenesis.** Point mutations of the $K_V7.1$ channel were engineered using overlap extension and high-fidelity PCR. Each mutation was verified by DNA sequencing. The cRNA of mutants was synthesized using the mMessage T7 polymerase kit (Applied Biosystems-Thermo Fisher Scientific). All primer sequences used in this study can be found in Supplementary table 1.

**Oocyte expression.** Stage V or VI oocytes were obtained from *Xenopus laevis* by laparotomy. All procedures were performed in accordance with the protocol approved by the Washington University Animal Studies Committee (Protocol # 20190030). Oocytes were digested by collagenase (0.5 mg/ml, Sigma Aldrich, St Louis, MO) and injected with channel cRNAs (Drummond Nanoject, Broomall).

Each oocyte was injected with cRNAs (9.2 ng) of WT or mutant $K_V7.1$, with or without KCNE cRNAs (2.3 ng). The amount of injected cRNAs was doubled for VCF experiments to achieve higher surface expression level. Injected cells were incubated in ND96 solution (in mM): 96 NaCl, 2 KCl, 1.8 $CaCl_2$, 1 $MgCl_2$, 5 HEPES, 2.5 $CH_3COCO_2Na$, 1:100 Pen-Strep, pH 7.6) at 18 °C for at least 2 days before recording.

**Western blot.** Western blot experiments follow standard protocol[35]. Primary antibody for KCNQ1 (conjugated with horseradish peroxidase HRP): KCNQ1 (G-8) mouse monoclonal IgG. Vendor: Santa Cruz. Cat. No.: sc-365186; primary antibody for Gβ: Gβ (T-20) rabbit polyclonal IgG. Vendor: Santa Cruz. Cat. No.: sc-378; secondary antibody for Gβ: Goat anti-Rabbit IgG (H+L), Vendor: Thermo. Cat. No.: A16110. Source data of western blot are provided in Source Data file.

**Electrophysiology experiments.** Microelectrodes (Sutter Instrument, Item #: B150-117-10) were made with a puller (Sutter Instrument, P-97), and the resistances were 0.5–3 MΩ when filled with 3 M KCl solution. Ionic currents were recorded by two-electrode voltage clamp (TEVC) in ND96 bath solutions. Whole-oocyte currents were recorded using a CA-1B amplifier (Dagan, Minneapolis, MN) with Patchmaster (HEKA) software. The currents were sampled at 1 kHz and low-pass-filtered at 2 kHz. All recordings were carried out at room temperature (21–23 °C). For voltage-clamp fluorometry (VCF) experiments, oocytes were incubated for 30 min on ice in 10 μM Alexa 488 C5-maleimide (Molecular Probes, Eugene, OR) in high $K^+$ solution in mM (98 KCl, 1.8 $CaCl_2$, 5 HEPES, pH 7.6) for labeling. Cells were washed three times with ND96 solution to remove the labeling solution, and recordings were performed in ND96 solution on the CA-1B amplifier setup. Excitation and emission lights were filtered by a FITC filter cube (Leica, Germany, for Alexa 488) and the fluorescence signals were collected by a Pin20A photodiode (OSI Optoelectronics). The signals were then amplified by an EPC10 (HEKA, analog filtered at 200 Hz, sampled at 1 kHz) patch clamp amplifier and controlled by the CA-1B amplifier to ensure fluorescence signals were recorded simultaneously with currents. All other chemicals were from Sigma Aldrich. Each electrophysiology experiment was performed on at least three individual cells to avoid possible outliers. These experiments are consistently reproducible as shown in the repeated recordings. The number of recordings of each experiment was based on the convention of the field.

**Electrophysiology data analysis.** Data were analyzed with IGOR (Wavemetrics, Lake Oswego, OR), Clampfit (Axon Instruments, Inc., Sunnyvale, CA), Sigmaplot (SPSS, Inc., San Jose, CA), and custom MATLAB (MathWorks, MA) software. The instantaneous tail currents following test pulses were normalized to the maximal current to calculate the conductance–voltage (G–V) relationship. Because of photo-bleaching, fluorescence signals were baseline subtracted by fitting and extrapolating the first 2 s signals at the −80 mV holding potential. ΔF/F was calculated after baseline subtraction. Fluorescence–voltage (F–V) relationships were derived by normalizing the ΔF/F value at the end of each four-seconds test pulse to the maximal value. F–V and G–V curves were fitted with either one or the sum of two Boltzmann equations in the form $G=1/(1+\exp(-z{*}F{*}(V-V_{1/2})/RT))$ where $z$ is the equivalent valence of the transition, $V_{1/2}$ is the voltage at which the transition is half maximal, $R$ is the gas constant, $T$ is the absolute temperature, $F$ is the Faraday constant, and $V$ is the voltage. For double mutant cycle analysis, the activation energy was given by $\Delta G=-z{*}F{*}V_{1/2}$. $F$ is the Faraday constant (23.06 kcal/mol). Both $z$ and $V_{1/2}$ were obtained by fitting the G–V relation with the Boltzmann equation[27,49–51].

**MD simulations conducted on $K_V7.1$ models.** The AO, IO, RC, and AC molecular models of $K_V7.1$ channel used for MD simulations were initially built by homology modeling[52] with MODELLER software, using the $K_V1.2$ channel refined crystal structure (PDB: 3LUT) as a template for the AO model[53]. For IO model, the γ intermediate conformation obtained in previous unbiased MD simulations of $K_V1.2$ refined structure in hyperpolarizing conditions[12] was used. For the RC model, the ε conformation obtained by biased MD simulations of $K_V1.2$ channel in hyperpolarizing conditions[12,54] was used as a template. The uncoupled $K_V7.1$ CryoEM structure[2] was used as a template for the AC model. These models were embedded in a 1-palmitoyl-2-oleoyl-phosphatidylcholine (POPC) bilayer and surrounded by two slabs of a 150 mM KCl solution. Considering the importance of phosphatidylinositol-4,5-bisphosphate ($PIP_2$) in $K_V7.1$ function[29,55], we incorporated a $PIP_2$ molecule at the bottom of each VSD, in the inner leaflet of the POPC bilayer for RC, IO and AO models, in accordance with previous MD simulations conducted on the $K_V7.1$ channel[6,56]. As the absence of the VSD-pore coupling in the $K_V7.1$ CryoEM structure is assumed to be due to the absence of $PIP_2$ lipids[2], we did not add this lipid in the MD system of AC model, in order not to induce any conformational change within the membrane. The MD simulations were performed using the NAMD program[57] along with the CHARMM force field[58,59]. All simulations were carried out in the NPT ensemble, so we applied Langevin dynamics to keep the temperature (300 K) and the pressure (1 atm) constant. The time evolution of the internal energy of our systems was monitored using the equations of motion on the atoms' cartesian coordinates. These equations were integrated with a time-step of 2 fs, using a multiple time-step algorithm in which

short and long-range forces were calculated every 1 and 2 time-steps, respectively. Long-range electrostatics were calculated using Particle Mesh Ewald. The cutoff distance of short-range electrostatics was set to 11 Å. A switching function was used starting at 8 Å to smoothly bring the Van der Waals (VdW) energies to 0 at 11 Å. During the calculations, chemical bond lengths between hydrogen and heavy atoms were maintained at their equilibrium values using SHAKE algorithm. Periodic boundary conditions were applied.

Simulations were conducted in four phases: the first one, of 2 ns, aimed at fully solvating the protein in the membrane, by setting up harmonic constraints on all the atoms of $K_V7.1$ and $PIP_2$, to let the lipid molecules rearrange themselves around the protein, and then to let the protein sidechains relax within the membrane by limiting the constraints to the protein backbone and $PIP_2$ phosphorus atoms. The 70 ns third step was conducted with restraints on the $K_V7.1$ backbone and without any constraints on $PIP_2$ atoms, to allow these lipids to rearrange around the protein complex and within the lipid bilayer. This step also had the aim of letting the density of the system reach a constant value. Finally, the last step corresponds to the so-called production phase. This step, performed without any specific forces on any coordinate of the system, lasted approximately 500 ns. Only the backbone of the selectivity filter (corresponding to the voltage-gated ion channel conserved TTIGYG sequence) was harmonically constrained, to prevent unexpected ion conduction during simulations. For all the trajectories, the RMSD of the protein from its initial structure reached a plateau at ~50 ns. The production phase trajectory was used for further analyses.

The analyses of resulting MD trajectories were conducted in two parts. The first part aimed at validating our structural models with respect to experimental restraints. Before investigating the molecular determinants of E–M coupling in the MD trajectories, all the relaxed MD models were validated rigorously against experimental data to ensure accuracy. In addition to the S2-S4 salt bridges (Supplementary Fig. 5a, b), interatomic distances were calculated between residue pairs previously identified by Cys scanning experiments[60] and double mutant cycle analyses[47,61]. Our MD models suggest that these residue pairs are prone to interact in all $K_V7.1$ models (Supplementary Fig. 5e, f). We also monitored the RMSD between the PD segments of $K_V7.1$ CryoEM and those of RC model throughout its MD trajectory and the average RMSD values remain below 2 Å, which supports the closed state conformation of our RC model (Supplementary Fig. 5g, h). The last step of structural validation of $K_V7.1$ models against experimental knowledge consisted in the calculation the average pore radii values of $K_V7.1$ models throughout their respective MD trajectory. During MD simulations, the backbone of selectivity filter residues was geometrically constrained throughout the production phase. In such calculations indeed, we do not study the ion diffusion through the channel (which would require holding a transmembrane potential), a process that is out of the scope of the paper. Qualifying a conformation as conductive refers here only to the ability of the gate to conduct, a property that we might estimate by analyzing the dimension of the gate or by determining the pore profile of the channel.

For each model, we measured the latter by calculating the average pore radii across the conduction pathway, using HOLE program[62]. In these analyses, the pore size of a given 3D channel structure is estimated by rolling a sphere within the conduction pathway[63]. The center of this sphere is adjusted by the Metropolis Monte Carlo simulated annealing procedure, in order to find the largest sphere. The final reported estimate of the pore radius is determined as that of this largest sphere.

The results we obtained (Supplementary Fig. 5c) present two major regions that discriminate open models against closed ones. The average pore radii obtained for RC and AC models are narrower than those of IO and AO models (Supplementary Fig. 5d), and indicate that closed state models are less likely to conduct potassium ions than open ones. In both directions along the pore axis, the radius of the sphere is maximized every 0.125 Å with respect to the VdW radius of the atoms facing the conduction pathway. HOLE program repeats this procedure throughout the whole pore until the sphere reaches a maximum radius of 5 Å.

Estimating the likelihood of $K_V7.1$ models to be conductive only considering their pore size is merely possible. Yet, the calculation of the average pore radii in the inner pore, located below the selectivity filter can be useful. Indeed, several studies shed light on the relation between the radius of a nanopore and the free energy associated to the translocation of an ion[64,65]. Similar results were obtained for the inner pore of $K_V1.2$ channel[66]. This study also highlighted the polarity of the atoms facing the pore as important for the free energy of K+ ion translocation, as polar atoms tend to decrease this free energy, while hydrophobic residues tend to increase it. In $K_V7.1$ CryoEM structure, the pore is constricted to 0.8 Å at the level of L353, a hydrophobic residue of S6 lining the inner pore, and located at −25 Å on the z axis (Supplementary Fig. 5d). In this regard, we compared the average pore radii obtained in this region in $K_V7.1$ models, and both AC and RC models present lower pore radii (0.9 Å and 1.4 Å, respectively) than IO and AO ones (4.0 Å and 5.0 Å, respectively). Overall, these results suggest that our $K_V7.1$ models are in good agreement with both experimental structures and site-directed mutagenesis studies, especially regarding their structure and their state-dependent interactions.

The second part of MD simulation analysis aimed at providing a molecular insight of intra- and inter-subunit interactions between key residues for the $K_V7.1$ E–M coupling. To achieve this goal, we selected, for each residue, the ones that remained at an average distance of 5 Å from its entire side chain in MD trajectories. For each resulting residue pair, we monitored the distance between their sidechains

in the four subunits of our models every 2 ns over the trajectory of the production phase. To characterize interactions commonly found in transmembrane proteins, such as Van Der Waals interactions[67], we computed distances between each terminal methyl group for aliphatic residues and recorded the shortest one. For pi-stacking interactions occurring between aromatic residues[68], we computed the distance between geometric centers of their respective aromatic rings. For these interactions, we used a distance cutoff of 5.1 Å. To characterize electrostatic interactions such as salt bridges or hydrogen bonds, we computed the distance between each charge group atom (or each heteroatom for hydrogen bonds) of the residue pairs involving charged or polar residues and recorded the shortest one, using a 3.5 Å distance cutoff. For the residue pairs including methionine residues, we characterized Met-Arom interactions[69,70] by computing the distance between methionine sulfur atoms and aromatic ring centers, using a distance cutoff of 7.1 Å. For Met-Leu interactions[70], we calculated the distance between Met sulfur atom, and the ramified carbon of Leu side chain, using a cutoff distance of 6.1 Å. Then, in order to provide a concise version of the data we obtained, we calculated, for each residue pair, the frequency at which the sidechains remain below the threshold values in all subunits throughout MD trajectories of each $K_V7.1$ model, depending on the nature of the interaction. Two residues of a pair were considered present if its frequency of interaction is above 0.6, which indicates that the interaction is present in at least three subunits over four. For residue pairs presenting frequency values inferior to 0.6 in all models, we will consider an interaction as present in a model if its frequency is at least 0.4 higher than in the model presenting the least frequency of interaction.

Figures representing $K_V7.1$ models were generated with Visual Molecular Dynamics (VMD) software[71]. All the analysis calculations were performed with Tcl programming scripts[72] and executed in VMD interface.

**Statistical coupling analysis (SCA)**. SCA was performed with the pySCA software package from the Ranganathan lab[45]. An underlying assumption of SCA is that amino acid co-evolution occurs to maintain interactions critical to protein functions. SCA analyzes amino acid positional co-evolution within protein families to identify protein sectors that may correspond to networks of amino acid interactions critical for protein function[44–46]. The input multiple sequence alignment (MSA) was derived from PF00520 (Ion_trans Family) MSA in the Pfam protein families database[73]. The full PF00520 MSA contains 58,529 ion transport protein sequences aligned from S1 through S6. All sequences in the PF00520 MSA were annotated using PFAM annotation based on the codes provided by the pySCA package[45]. The annotations were used to filter and select sequences for SCA. Inclusion keywords annotation were voltage-gated or potassium to select for all $K_V$ and potassium channels. Exclusion keywords were unknown, uncharacterized, slo, Calcium-activated, cyclic, subfamily H, and eag to exclude all uncharacterized protein and non-domain-swapped $K_V$ channels. This filtering on PF00520 resulted in our domain-swapped $K_V$ preliminary input MSA with 1804 $K_V$ sequences at 2225 positions which included members from $K_V1 – K_V9$, and did not include members from $K_V10$, $K_V11$, or $K_V12$. Our input MSA is distinct from a prior SCA computed on the $K_V$ superfamily[46], in which $K_V$ channels featuring the two distinct architecture (domain-swapped and non-domain-swapped) were combined as input for SCA analysis. The two architectures of $K_V$ channels likely participate in distinct E–M coupling mechanisms. Our SCA analysis aims to extract potential E–M coupling networks specifically within the domain-swapped $K_V$ channels.

The preliminary input MSA filtered from PF00520 underwent further processing by pySCA with default parameters[45] and the reference sequence set to the human $K_V7.1$ sequence. To briefly reiterate the default parameters: gapped positions were truncated with a gap fraction cutoff of 0.2 based on position and sequence. Sequences with <0.2 fractional identity to human $K_V7.1$ were removed from analysis. Sequence weights were computed with max fractional identity of 0.8 to the reference sequence. The final processed MSA was truncated based on the available residues in a human $K_V7.1$ homology model of the *Xenopus* $K_V7.1$ cryoEM structure[2], resulting in a final MSA size of 1,421 sequences at 200 positions (Fig. 6a, Supplementary Fig. 7a). This final MSA yielded first order conservation consistent with the topology of $K_V$ channels (Supplementary Fig. 7b). First order conservation was calculated as defaulted in pySCA[45]. Briefly, first order conservation was calculated with Kullback-Leibler divergence which describes deviation of the observed amino acid frequency within the MSA against the pre-defined natural background amino acid frequency.

Protein sectors were defined based on the IC-based submatrix derived from the SCA matrix (Fig. 6b) utilizing pySCA[45]. SCA first calculated the SCA covariance matrix between all pairwise positions within the final MSA (Fig. 6b). Next, spectral analysis was applied to the covariance matrix to extract top eigenmodes corresponding to top co-varying residue positions (Supplementary Fig. 7c). Significant top eigenmodes from the SCA covariance matrix was determined by comparing the results of spectral analysis of the SCA covariance matrix (Supplementary Fig. 7c, black) with 10 trials of randomized input (Supplementary Fig. 7c, red). This comparison identified five top eigenmodes within the SCA covariance matrix (Supplementary Fig. 7c, blue arrows). Next, SCA transformed the selected top eigenmodes into independent components (ICs) by utilizing independent component analysis. Independent component analysis rotated the five eigenmodes to five ICs such that each IC contains residue positions which most strongly co-vary among themselves and most weakly co-vary with residue positions

in other ICs. Key residues in each independent component (IC) were determined by top five percent of the cumulative density function of each IC after independent component analysis (default for pySCA). The five identified ICs are shown in the IC submatrix and mapped onto the KCNQ1 cryo-EM structure (Fig. 6b, Supplementary Fig. 7d), corresponding to top five sets of co-evolving amino acid residue positions in the domain-swapped $K_V$ family. In the IC submatrix, on-diagonal boxes report the strength of internal correlations for residues within an IC; while the off-diagonal boxes show the strength of external correlations for residues between two ICs[45]. Strong external correlations between ICs indicate that these ICs can be grouped into a single protein sector[45]. As an example in our analysis, IC3 featured strong external correlation with IC2, but weak external correlation with IC1, IC4 or IC5 (Fig. 6b, row 3), indicating that IC2 and IC3 together form one protein sector. The five detected ICs can be roughly grouped into two sectors: (1) IC2+IC3 and (2) IC1+IC4+IC5 (Fig. 6b). We accordingly defined two protein sectors within the domain-swapped $K_V$ families by grouping the two ICs (Fig. 6b–e).

**Homology models of h$K_V$7.1 and molecular docking**. There are two templates available for modeling human $K_V$7.1 (h$K_V$7.1): the cryo-EM structure of frog $K_V$7.1 (PDB entry: 5vms)[2] and the crystal structure of rat $K_V$1.2-$K_V$2.1 chimera (PDB entry: 2r9r)[74]. These two template structures have significantly different conformations, particularly in the regions near the S4-S5L and the VSD-pore interface. Therefore, both templates were used, and two h$K_V$7.1 structures (5vms_model and 2r9r_model) were built using the program MODELLER[75]. Both homology models of h$K_V$7.1 were used in the following molecular docking studies.

The compound ML277 was docked to each h$K_V$7.1 model structure using our previously developed method[76–78]. Briefly, a modified version of AutoDock Vina[79] was employed to sample possible binding modes of ML277 on h$K_V$7.1. In this modified version, the maximum number of output modes is user-specified, and was set to 500 in the present study. The exhaustiveness value was increased to 30 to ensure exhaustive sampling. The protein was treated as a rigid body, and the single bonds in ML277 were considered as rotatable. The docking site was focused on the pocket near the S4-S5L and the VSD-pore interface based on the experimental data. The size of the cubic search box was set to 30 Å, which was sufficiently large to cover the whole binding pocket. Up to 500 putative, flexible binding modes were generated and then re-ranked with our in-house scoring function, ITScore[76,77]. The predictions based on the 2r9r_model (i.e., the homology model of h$K_V$7.1 that was constructed using 2r9r as the template) achieved significantly better binding scores than the 5vms_model (i.e., the homology model of h$K_V$7.1 based on 5vms). Therefore, we focused on the 2r9r_model; the predicted binding mode is plotted in Supplementary Fig. 1, and is consistent with the mutagenesis data. Interacting residues are represented by both the stick model and the surface representation.

**Reporting summary**. Further information on research design is available in the Nature Research Reporting Summary linked to this article.

## Data availability
Data supporting the findings of this manuscript are available from the corresponding author upon reasonable request. A reporting summary for this Article is available as a Supplementary Information file. The source data for Figs. 1a,c,e–g, 2a–d, 3b–g, 4a, 5a–c, sFigs. 2, 3, MD trajectories, and code for SCA are provided in the Source Data file.

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

## Acknowledgements

The authors thank Prof. Jeanne Nerbonne for helpful discussion. This work was supported by R01 NS092570 and R01 HL126774 to J.C., by R01 GM019980 to X.Z., and by AHA 18POST34030203 to P.H.

## Author contributions

J.C., M.T. and P.H. conceived this work and directed the approaches used. The experimental work of this paper was conducted by P.H., N.-D.Y., J.S., Y.L., K.M.W., G.S. and L.Z. The MD simulation work was conducted by A.D.K., M.A.K. and M.T. The SCA was done by P.W.K. and M.A.Z. The ML277 docking results were done by X.X. and X.Z. All authors participated in data analysis. J.C., M.T., P.H., P.W.K. and A.D.K. wrote the paper with input from all authors.

## Competing interests

J.S. and J.C. are cofounders of a startup company VivoCor LLC, which is targeting $I_{Ks}$ for the treatment of cardiac arrhythmia. Other authors declare they have no competing interests.
