## [Peer Review File · Nature Communications]

Reviewers' Comments:

Reviewer #1:

Remarks to the Author:

The authors approached the EM coupling of KCNQ1, using various effective techniques such as electro- physiology and VCF with systematic mutagenesis, MD simulation and also MSA analysis. They observed the classic intra- subunit interaction between S4-S5L (hand) and S6c in both IO and AO state. They also newly identified another inter-subunit interaction of S4/S4-S5L joint (elbow) with S5/S6 on the neighboring subunit, only in the AO state.

I judge the elucidation of dynamic molecular mechanisms of two-stage EM coupling is important, as it cannot be achieved solely by structure analysis. I also judge the statistical coupling analysis of other numerous domain- swapped Kv channels using MSA analysis elevated the scientific merit of the work. This paper is written well and will attract attention of wide-ranged readers.

I have some specific comments which require attention.

1. Fig. 5e

In the scheme, the hand interaction between S4-S5L and the pore occurs not in the RC state but in the IO state. Also, it is described in the text as follows.

(lines 271-273) "The S4 (upper arm) movement into the intermediate state triggers the C-terminus of S4-S5L(hand) to grip the S6c of the same subunit to promote channel opening at the first conductive IO state."

(lines 308-309) "the classic E-M coupling interactions at S4-S5L/S6c engage upon VSD movement into the intermediate state".

However, the results of MD simulation in Fig. 5a clearly show that the interactions (e.g. between V254 and L347) occur equally well both in the RC state and IO state. I would like to request full explanation and revisions to clarify this apparent discrepancy.

2. Fig. 5b

I understand this is the IO state model. I would like to see interactions depicted on the structure in the AO state, as well as in the RC state. These interactions is expected to be seen also in the AO and RC states, judging from the data in Fig. 5a.

3. Fig.5

The results of MD analysis is thought to be strongly influenced by the structure model in each state. Thus, the validity of the structure model is very critical. By the detail of the structure model in the Methods section (lines 533-540), the AO and AC models look to be convincing, but I wonder if the IO and RC models are trustable enough.

4. Fig. 2a

The kinetics of current activation of W248R is very slow, although it is thought to be in the IO state. Is there any reasonable explanation?

5. Fig. 2d

Current amplitude scale (and also time scale) is missing.

6. I understand this is not the major focus of this work, but I wonder if the mechanistic insight in the present work can explain the reason why IO does not happen in KCNQ1/KCNE1 complex.

Reviewer #2:

Remarks to the Author:

The authors here propose a two-stage "hand-and-elbow" gating mechanism for KCNQ1 channel. In voltage-gated K channels, the channel opening has been proposed to be coupled to the voltage-

sensing domain (VSD) movement via an electro-mechanical (E-M) coupling using the S4-S5 linker as a coupling domain. In KCNQ1 channels, the VSD activates in two steps: from a resting state to an intermediate-activated state and then to the fully-activated state. The authors first identified mutations that prevent both intermediate opening and fully activated opening. These residues were consistent with the classical E-M coupling. In an additional screen using ML277 to identify mutations that exclusively prevents the ML-277-induced increase in current, the authors proposed that they identified residues important for fully-activated opening. Using four different tests, all of these residues seem important for the fully-activated opening. These residues did not seem to fall into the classical E-M coupling area, but mapped onto the interface between subunits. They further tested some of these pairs of interactions between the S4-S5 linker and the pore in neighboring subunits using Double Mutant Cycle analysis. These new activated-open E-M coupling interactions identified in the study is novel. The authors then performed MD simulation that support their idea that the classic interactions are important in the IO and AO E-M coupling while the newly-identified AO E-M coupling contribute to opening only in the fully-activated state. The authors concludes that when S4 moves to intermediate state, the hand-like C-terminus of S4-S5 linker interact with the pore in the same subunit and thereby promotes opening. This is consistent with the classic E-M coupling in many studies. When S4 then moves to fully-activated state, the elbow-like hinge between S4 and S4-S5 linker also interact with the neighboring pore domain to further activate the conductance. By using Statistical Coupling Analysis, they also suggest that the two-state "hand-and-elbow" model could be applicable to many Kv channels. This newly proposed "hand-and-elbow" mechanism helps to understand how VSD movement is coupled to channel opening and to explain how some mutations change channel function and thus lead to cardiac diseases such as Long QT syndromes. In addition, as different auxiliary subunits (eg. KCNE1 and KCNE3) modulate KCNQ1 gating by changing the E-M coupling in both VSD states, the mechanism provides new insights into the modulation of KCNQ1 by tissue-specific auxiliary subunits. The data are convincing, exhaustive, and robust, and the conclusion are appropriate. However, I would like some additional discussion and more detailed explanations of some figures (see below).

Major comments.

1. Pg. 6. Line 129. The authors identified some mutations featuring strong fluorescence signals and no detectable currents, which could suggest that those residues are important for the classic E-M coupling. But the data presented can't directly suggest any interactions between them. Thus, "these results suggest that in Kv7.1 the E-M coupling interactions are also necessary for pore opening" seems too strong. Could some of these mutations just be important for gate opening, not coupling per se (e.g. A341V, P343A)?
2. Pg. 4. Line 68. "Kv7.1 is the only..." this statement seems too strong. Some studies have shown that KCNQ1 shows one open state (Kass et al 2010, Kubo et al 2014). Also, the GV of Kv7.1 has two components or not? Could it be fitted by double Boltzmann function as FV does? Maybe show the two components in Fig 1?
3. Fig. 2a Show fluorescence from W248R+KCNE1 to show that the channels are expressing and that mutations did not prevent S4 movement, which are alternative explanations for no currents. Please also add the fluorescence in presence of KCNE1 for as many as possible of the mutations in Supplementary Fig 2 to show the same. Understandably, it might not work for all of them, but at least for a couple of them.
4. I have some trouble with the 5-state model. If a mutation prevents AO state, like W248R, then should not current go to 0 at very depolarized voltages (i.e. channel will be pushed into AC by the positive voltage)? Similarly, for a mutation like L251A, which also is supposed to prevent AO state, why are there still two components in the current time course? Previously, the two components were interpreted as IO and AO activation.
5. Fig. 5. The simulation and the experimental data are not always using the same interacting pairs (e.g. Fig. 5c). Please explain and make it clear to the reader when data and simulations match and when they don't match.
6. Fig 5c bottom. For interacting pairs that are less prevalent in the AO state, shouldn't mutations make it easier to reach AO state (i.e. interactions hold the channel in RC or IO states and mutations should make these state less stable)?

7. Fig. 6. The SCA analysis cannot distinguish residue-residue interactions important for coupling compared to e.g. gate opening. Maybe point out some residue-residue interactions in Fig 5e that were found here to be important for coupling?

Minor Comments.

1. Pg. 3. Line 60. There seems to be a gap in discussion of the cardiac disorders and E-M coupling here. A little bit more introduction about the E-M coupling causing cardiac disorders will be appreciated.
2. Fig 1b. Compared to a previous gating model of Kv7.1 where there were six states, including resting-open state (Hou et al 2017), why here RO state is missing?
3. Pg. 8. Line 170. Why is exactly the region of the classical E-M coupling? Conclusion that "W248 and S338 are located outside of the region of the classical E-M coupling" seems too strong. And are V255 and F256 not part of classical region, but V254 and H258 are?
4. Pg. 2. Line 32. "Here, we leverage" this sentence is too complicated to understand for readers.
5. Pg. 2. Line 36. Please mention that the novel AO state E-M coupling interactions are from neighboring subunits. Otherwise, readers might get confused.
6. Fig. 2a. Why so few traces in KCNQ1 W248R currents, when GV shows many points with currents?
7. Fig 2e and 2h. The V50 of G-V is missing or too small in both figures.
8. Pg. 10. Line 232. How to get the activation energies of voltage-dependent action of channels should be explained.
9. Pg. 14. Line 339 and Pg. 30. Line 604. "K+" should be "K⁺".
10. Fig 3c. Missing bar for Kv7.1 in top row.
11. Fig 5e. If possible, color the subunits with another color instead of grey. E.g., it's hard to see the label "S4-S5 linker" in the figure.
12. Supplementary Fig3. R243W GV curve shows robust constitutive current. But the current traces don't show any constitutive current. Why is that?
13. Supplementary Fig.4 line 52. Analyses should be "analysis"
14. Supplementary Fig 5b and e. I would remove the column marked Assigned State. To me that is just confusing.
15. Supplementary Fig 5e. Shouldn't F232 move relative to F279 when S4 moves up? One would not expect these two residues to always be together in RC, IO, and AO?
16. Supplementary Fig. 6b. Please explain y axis scale? E.g., 3.5 means 10% conserved or 100% conserved? In addition, shouldn't GYG be the most conserved in a K channel alignment?
17. Supplementary Fig. 6c. Show the data with arrows better (e.g. Y axis log scale, or inset with finer scale).
18. Legend Supplementary Fig. 6d. References to Fig 4c, d, and e should be Fig 6c, d, and e.

Reviewer #3:

Remarks to the Author:

In this paper by Hou et al., the authors proposes a two-stage "hand-end-elbow" gating mechanism of Kv7.1 voltage-gated potassium channel. The authors used several experimental and simulation techniques to reach this conclusion. Overall, the manuscript is well written and worthy of publication pending minor to moderate revisions. I have several comments to further improve the manuscript:

1. The SCA analysis is carried out only on domain-swapped Kv sequences (for good reasons) and is clearly mentioned in the results and the methods sections. However, the abstract reads differently and sounds like this mechanism is applicable to any Kv channel (Page 2, Line 39 and 40). Would not it be better to modify "numerous Kv channels" to "numerous Kv channels with domain-swapped architecture"? We have plenty of evidence now that non-swapped topology channels are gating very differently and soluble/regulatory domains playing very significant role in the process, see for example:

Perissinotti, Laura L., et al. "Determinants of isoform-specific gating kinetics of hERG1 channel: Combined experimental and simulation study." *Frontiers in physiology* 9 (2018): 207
Barros, Francisco, Pedro Domínguez, and Pilar de la Peña. "Relative positioning of Kv11. 1 (hERG) K⁺ channel cytoplasmic domain-located fluorescent tags toward the plasma membrane." *Scientific reports* 8.1 (2018): 15494.

and more recent:

Whicher, J. R., & MacKinnon, R. (2019). Regulation of Eag1 gating by its intracellular domains. *eLife*, 8.

I do feel that emphasizing apparent differences between two topologies present in Kv family may help a lot with application domain of findings reported and will place it into a broader context.

2. Page 17, Line 385: From the text, it seems the cell line measurements are repeated more than 3 times. Can the authors provide further clarification on the reliability of these measurements? Is it reliable and if so, why? It will be helpful for inexperienced reader if this is discussed further in the methods section.

3. Page 28, Line 548: The reference for CHARMM force field is wrong. The authors mentioned reference is for CHARMM simulation program. Please provide the correct references for CHARMM protein, lipids, ions etc. Please also provide the CHARMM force field versions used.

4. Page 28, Line 556: I suppose the authors used SHAKE algorithm. Please mention that.

5. Page 29, Line 567: Do the authors really mean "spatially constrained"? This might have unnecessary effect on the other parts of the system during production run, in my experience. More reasonable choice would be to use restraints.

Reviewers' comments:

Reviewer #1 (Remarks to the Author):

The authors approached the EM coupling of KCNQ1, using various effective techniques such as electro-physiology and VCF with systematic mutagenesis, MD simulation and also MSA analysis.

They observed the classic intra- subunit interaction between S4-S5L (hand) and S6c in both IO and AO state. They also newly identified another inter-subunit interaction of S4/S4-S5L joint (elbow) with S5/S6 on the neighboring subunit, only in the AO state.

I judge the elucidation of dynamic molecular mechanisms of two-stage EM coupling is important, as it cannot be achieved solely by structure analysis. I also judge the statistical coupling analysis of other numerous domain- swapped Kv channels using MSA analysis elevated the scientific merit of the work. This paper is written well and will attract attention of wide-ranged readers.

We thank the reviewer for the positive and encouraging evaluation.

I have some specific comments which require attention.

1. Fig. 5e

In the scheme, the hand interaction between S4-S5L and the pore occurs not in the RC state but in the IO state. Also, it is described in the text as follows.

(lines 271-273) "The S4 (upper arm) movement into the intermediate state triggers the C-terminus of S4-S5L(hand) to grip the S6c of the same subunit to promote channel opening at the first conductive IO state."

(lines 308-309) "the classic E-M coupling interactions at S4-S5L/S6c engage upon VSD movement into the intermediate state".

However, the results of MD simulation in Fig. 5a clearly show that the interactions (e.g. between V254 and L347) occur equally well both in the RC state and IO state. I would like to request full explanation and revisions to clarify this apparent discrepancy.

We thank the reviewer for raising this good point. The results of MD simulation in Fig. 5a show that interactions involved in the classic E-M coupling "occur equally well" at different states during the channel gating including RC, AC, IO, and AO. These results suggest that even when the VSD is at the resting state: 1) these pairs of residues stay close to each other and the interactions may be already engaged; 2) these interactions may promote channel opening upon the VSD activation to intermediate state and remain effective when the VSD moves to activated state. We have now revised this important point to "the motion of the S4 promotes channel opening through the S4-S5L (hand) grip of the S6c of the same subunit at the first conductive IO state" (pg. 12) and "the classic E-M coupling interactions at S4-S5L/S6c promotes channel opening upon VSD movement into the intermediate state" (pg. 13), we have also revised the cartoon scheme model in Fig. 5e to show the "hand grip" at RC state.

2. Fig. 5b

I understand this is the IO state model. I would like to see interactions depicted on the structure in the AO state, as well as in the RC state. These interactions is expected to be seen also in the AO and RC states, judging from the data in Fig. 5a.

We have prepared a new figure showing the classical E-M interactions at AO and RC states, as the reviewer expected, these interactions also present in AO and RC states (rFig. 1). We have added this as a supplementary figure 6.

rFig. 1 Possible intra-subunit interactions at the classical E-M coupling region in RC and AO states. Representations of the IO and AO state models of a Kv7.1 subunit. Inset: enlarged structures of the boxed area. Blue, residues identified as important for E-M coupling in experiments (Fig. 1c-g); grey: V255 and T265, which were identified in MD simulations as important for E-M coupling, but for which mutations still show functional currents (Fig. 3c). Residue V255 was found to be important for the AO state E-M coupling (Fig. 3). Each pair of residues is represented by a black dashed line connecting their respective sidechains.

3. Fig.5

The results of MD analysis is thought to be strongly influenced by the structure model in each state. Thus, the validity of the structure model is very critical. By the detail of the structure model in the Methods section (lines 533-540), the AO and AC models look to be convincing, but I wonder if the IO and RC models are trustable enough.

The reviewer makes a good point. We have made different efforts to improve the accuracy of our models at all different states. 1) The VSD salt-bridges of these models (Supplementary Fig. 5a,b) are satisfying previous functional data (Wu et al. BJ 2010; Zaydman et al. eLife 2014). 2) The intermediate state VSD is also supported by additional electrophysiology data and NMR intermediate state structure of Kv7.1 (PDB ID: 6MIE, provided as related manuscript for review, also see Taylor et al. Biophysical Journal 2019), and supported by the results that the average RMSD values remained below 5 Å throughout IO model MD trajectory. 3) We also monitored

rFig. 2 Comparison of the pore domain of Kv7.1 RC model with KCNQ1_{EM} structure.

On the left panel, the graph project the root-mean-square deviation (RMSD) of the PD segments of Kv7.1 RC model with respect to KCNQ1_{EM} structure throughout its MD simulation time (in ns). Average RMSD values are displayed for each subunit (labeled chain A, B, C, D). The orange vertical line represents the simulation time at which the pictures displayed on the right panels were captured. The right panels show a cartoon representation of the structural alignment of the S6 segment of KCNQ1_{EM} structure (in transparent orange) with those of Kv7.1 (top) RC subunits, each captured at ~420 ns. Each subunit is represented in solid black, red, green and blue cartoon, respectively.

the RMSD between the PD segments of Kv7.1 CryoEM and those of RC model throughout its MD trajectory and the average RMSD values remain below 2 Å, which supports the closed state conformation of our RC model (rFig. 2).

We have added rFig. 2 to supplemental Fig. 5 (panel g, h), and cited the Figure in the main text (page 33).

4. Fig. 2a

The kinetics of current activation of W248R is very slow, although it is thought to be in the IO state. Is there any reasonable explanation?

This is a good observation by the reviewer. W248R demonstrated consistent read-out across all four distinct tests (KCNE1 co-expression; co-mutation with E1R/R4E, E1R/R2E, and F351A) to show that it selectively opens in the IO state. The mutation may change the conformation of the channel in a complex way that also slows down certain transitions during activation to the IO state, which we did not investigate in this study.

5. Fig. 2d

Current amplitude scale (and also time scale) is missing.

Corrected.

6. I understand this is not the major focus of this work, but I wonder if the mechanistic insight in the present work can explain the reason why IO does not happen in KCNQ1/KCNE1 complex.

This is an important question, but unfortunately, there is not sufficient structural or functional data to address this question at this time.

Reviewer #2 (Remarks to the Author):

The authors here propose a two-stage “hand-and-elbow” gating mechanism for KCNQ1 channel. In voltage-gated K channels, the channel opening has been proposed to be coupled to the voltage-sensing domain (VSD) movement via an electro-mechanical (E-M) coupling using the S4-S5 linker as a coupling domain. In KCNQ1 channels, the VSD activates in two steps: from a resting state to an intermediate-activated state and then to the fully-activated state. The authors first identified mutations that prevent both intermediate opening and fully activated opening. These residues were consistent with the classical E-M coupling. In an additional screen using ML277 to identify mutations that exclusively prevents the ML-277-induced increase in current, the authors proposed that they identified residues important for fully-activated opening. Using four different tests, all of these residues seem important for the fully-activated opening. These residues did not seem to fall into the classical E-M coupling area, but mapped onto the interface between subunits. They further tested some of these pairs of interactions between the S4-S5 linker and the pore in neighboring subunits using Double Mutant Cycle analysis. These new activated-open E-M coupling interactions identified in the study is novel. The authors then performed MD simulation that support their idea that the classic interactions are important in the IO and AO E-M coupling while the newly-identified AO E-M coupling contribute to opening only in the fully-activated state. The authors concludes that when S4 moves to intermediate state, the hand-like C-terminus of S4-S5 linker interact with the pore in the same subunit and thereby promotes opening. This is consistent with the classic E-M coupling in many studies. When S4 then moves to fully-activated state, the elbow-like hinge between S4 and S4-S5 linker also interact with the neighboring pore domain to further activate the conductance. By using

Statistical Coupling Analysis, they also suggest that the two-state “hand-and-elbow” model could be applicable to many Kv channels. This newly proposed “hand-and-elbow” mechanism helps to understand how VSD movement is coupled to channel opening and to explain how some mutations change channel function and thus lead to cardiac diseases such as Long QT syndromes. In addition, as different auxiliary subunits (eg. KCNE1 and KCNE3) modulate KCNQ1 gating by changing the E-M coupling in both VSD states, the mechanism provides new insights into the modulation of KCNQ1 by tissue-specific auxiliary subunits. The data are convincing, exhaustive, and robust, and the conclusion are appropriate. However, I would like some additional discussion and more detailed explanations of some figures (see below).

We thank the reviewer for the thorough and positive evaluation.

Major comments.

1. Pg. 6. Line 129. The authors identified some mutations featuring strong fluorescence signals and no detectable currents, which could suggest that those residues are important for the classic E-M coupling. But the data presented can't directly suggest any interactions between them. Thus, “these results suggest that in Kv7.1 the E-M coupling interactions are also necessary for pore opening” seems too strong. Could some of these mutations just be important for gate opening, not coupling per se (e.g. A341V, P343A)?

We appreciate that the reviewer raised a very important issue. We think that the residues in our study that their mutations get rid of the current but show strong fluorescence signal are important for E-M coupling. For residues (V254M and H258W) in the S4-S5L, since they are not in the pore and yet they affect the pore opening, they should be naturally thought to be important for the E-M coupling. On the other hand, residues (A341V, P343A, and G345A) in the S6 segment might be directly affecting pore opening, as the reviewer pointed out. However, the following evidence suggests that they also are important for the E-M coupling. 1) They are close to those residues important for the coupling in the S4-S5L, and MD simulation suggests that these residues interact (Fig. 5a). 2) These residues in Kv7.1 are hydrophobic and show similar spatial location with the classic E-M coupling residues identified in other Kv channels [Lu et al. Nature, 2001; Long et al. Science, 2005]. To this point, it makes sense that a residue in the pore might be important for the E-M coupling to sense the signal from the VSD and also important for pore opening. Therefore, it is hard to define clearly whether such residues in the pore directly affect the pore opening or coupling. With these considerations, we modified the sentence to “these results suggest that in Kv7.1 the E-M coupling interactions may also be necessary for pore opening”.

2. Pg. 4. Line 68. “Kv7.1 is the only...” this statement seems too strong. Some studies have shown that KCNQ1 shows one open state (Kass et al 2010, Kubo et al 2014). Also, the GV of Kv7.1 has two components or not? Could it be fitted by double Boltzmann function as FV does? Maybe show the two components in Fig 1?

It seems to us that this question is about whether Kv7.1 has two open states. In our previous studies (Zaydman et al. eLife 2017; Hou et al. Nat Commun 2017; Hou et al. eLife 2019), we have provided many lines of evidence to show the two open states. In this study, we also provide evidence in Figs 1-3 for the two open states.

The opening of the Kv7.1 channel also shows two components which are manifested as activation and a fast inactivation (Hou et al. Nat Commun 2017). The G–V relation of the KCNQ1 channel often shows a decay at high voltages above +50 mV, which could be the second component due to a reduced open probability when the VSD moves from the intermediate state to the activated state (Hou et al. Nat Commun 2019). However, in this high

voltage, some endogenous channels may also open, making it difficult to accurately show the second component.

3. Fig. 2a Show fluorescence from W248R+KCNE1 to show that the channels are expressing and that mutations did not prevent S4 movement, which are alternative explanations for no currents. Please also add the fluorescence in presence of KCNE1 for as many as possible of the mutations in Supplementary Fig 2 to show the same. Understandably, it might not work for all of them, but at least for a couple of them.

We thank the reviewer for the constructive suggestion. Using western blot and imaging data, Hoosien et al. have demonstrated that S338F+KCNE1 showed normal membrane expression although no current was detected (Hoosien et al. Heart Rhythm. 2013).

We performed new VCF experiments and obtained results of S338F+KCNE1 and L251A+KCNE1, which show clear voltage-dependent fluorescence signals. The reviewer is correct that the experiments did not work for other mutations with several attempts but failing to obtain stable fluorescence signals. It is well known that the association of KCNE1 makes the VCF experiments more difficult, and we only managed to get the results for two of these mutations.

We have added the new VCF results of S338F+KCNE1 and L251A+KCNE1 into the manuscript (pg. 9) and Figure S2.

4. I have some trouble with the 5-state model. If a mutation prevents AO state, like W248R, then should not current go to 0 at very depolarized voltages (i.e. channel will be pushed into AC by the positive voltage)? Similarly, for a mutation like L251A, which also is supposed to prevent AO state, why are there still two components in the current time course? Previously, the two components were interpreted as IO and AO activation.

These are good points. We have used the same model in previous studies (Hou et al. Nat Commun 2017; Hou et al. eLife 2019), and based on our simulation to experimental results, we obtained parameters for the model. The simulations fit data well, which did not show that the currents go to 0 at our tested voltages to 60 mV.

Mutations may cause complex conformational changes which not only ablate one of the open states but also affect other transitions. At this time, we do not know whether and how L251A affects other transitions that lead to multiple kinetic components.

5. Fig. 5. The simulation and the experimental data are not always using the same interacting pairs (e.g. Fig. 5c). Please explain and make it clear to the reader when data and simulations match and when they don't match.

For experimental data, using double mutant cycle in Fig. 4, we focused on determining inter-subunit interactions specifically responsible for the AO state. All residue pairs were chosen from those 17 residues (13 residues from the ML277 effect screening and 4 S4c residues) that have been demonstrated as important for the AO state E-M coupling. Our MD simulation data match well with these experimental data. In addition to those experimentally demonstrated interactions, we also determined possible nearby interacting residue pairs that include at least one of the 17 residues. These interactions fall into three groups of state-dependent interactions as shown in Fig. 5c,d. For residues V241 and F275, which were identified in MD simulations as possibly involved in sidechain interactions but for which mutations did not alter AO states, we were using dark grey color to distinguish them. These similarities and differences were described in the legend of Fig. 5d.

6. Fig 5c bottom. For interacting pairs that are less prevalent in the AO state, shouldn't mutations make it easier to reach AO state (i.e. interactions hold the channel in RC or IO states and mutations should make these state less stable)?

For interacting pairs that are less prevalent in the AO state, these interactions are present in the RC and IO states, and may need to be broken when the VSD activates to the fully activated state to enable AO state E-M coupling. The interaction provides difference in free energy for the transition between AO and other states. When the interaction is altered by the mutations, the experimental results suggest that the free energy differences no longer favor the AO state. It is hard to predict whether a mutation should stabilize or destabilize the AO state simply based on the interaction at a single state.

7. Fig. 6. The SCA analysis cannot distinguish residue-residue interactions important for coupling compared to e.g. gate opening. Maybe point out some residue-residue interactions in Fig 5e that were found here to be important for coupling?

Reviewer 2 makes an excellent point. SCA detects amino acid co-evolution within a sector, which strongly suggests these residues form a functional unit within the protein. However, SCA does not specify the particular function conferred by each sector. We defined sector 2 in our study as an "E-M coupling" sector by its spatial pattern. It is possible that sector 2 also contains residues important in VSD activation and gate opening; however, the large number of residues in the S4-S5 linker strongly suggests that sector 2 includes significant E-M coupling residues. We did not expect to explicitly match SCA sector residues with KCNQ1 AO state E-M coupling residues. This is because SCA outputs protein sectors of the "average" domain-swapped KV channel derived from the sequence alignment. The positions within each sector thus represent important positions "on average". In this light, complete matching between SCA E-M coupling sector and KCNQ1 coupling residues would only be expected if KCNQ1 is an exact "average" KV channel – this is not a claim we make. Our interpretation of the SCA result is that the "average" domain-swapped K_V channel features a group of co-evolving residues that are spatially reminiscent and compatible with the two-stage E-M coupling process we elucidated in KCNQ1, suggesting that our findings may broadly translate to other domain-swapped KV channels.

To more directly address the reviewer's question, we indeed find that SCA identified interactions we found to be important for AO state EM coupling in KCNQ1. For example, W248-I268 and L251-I268 were found to interact during KCNQ1 activation by mutant cycle (Fig. 4c) and in SCA sector 2 (Fig. 6e). Of the 13 AO state E-M coupling residues identified by our ML277 screen, SCA placed 6 residues (W248, L251, V255, I268, S338, and L342) into sector 2 (E-M coupling sector), while the remaining 7 residues were not categorized into sector 1 or 2. Quantitatively, sector 2 (E-M coupling sector) contains 50 residues out of 200 residues analyzed. If SCA identified E-M coupling residues by random chance, we would expect a 50/200 or 25% positive rate. Reassuringly, SCA picked up 6/13 E-M coupling residues we identified in KCNQ1 for a 46% positive rate, a significant enrichment over random chance. Moreover, we interpreted sector 1 as residues important for independent stabilities/functions of VSD and Pore and not involved in E-M coupling. Sector 1 contains 42 residues, but none of the AO-state E-M coupling residues identified in our screen, which represent a significant negative enrichment over random chance. The combination of the negative/positive enrichment in the SCA sectors 1 and 2 when compared to our ML277 screen lends confidence that our interpretation of the SCA sectors.

We have modified the SCA paragraph in the discussion section to address the important point Reviewer 2 raised in that sector 2 may include residues not exclusive to E-M coupling.

Minor Comments.

1. Pg. 3. Line 60. There seems to be a gap in discussion of the cardiac disorders and E-M coupling here. A little bit more introduction about the E-M coupling causing cardiac disorders will be appreciated.

We appreciate the constructive suggestion, and have added sentences discussing the link between E-M coupling and cardiac disorders (pg. 3).

2. Fig 1b. Compared to a previous gating model of Kv7.1 where there were six states, including resting-open state (Hou et al 2017), why here RO state is missing?

In addition to IO and AO states, the KCNQ1 channel has a very small open probability when the VSD is at the resting state (Ma et al. Biophys. J. 2011). This resting open (RO) state was included in the six-state KCNQ1 model (Hou et al. Nat Commun 2017). Based on experimental data and the six-state KCNQ1 model, the probability of transition into this RO state is very small, such that removing it changes little of other transitions. Removing the RO state also helps understand the two-stage “hand-and-elbow” mechanism.

3. Pg. 8. Line 170. Why is exactly the region of the classical E-M coupling? Conclusion that “W248 and S338 are located outside of the region of the classical E-M coupling” seems too strong. And are V255 and F256 not part of classical region, but V254 and H258 are?

This is a really good point. There can be some spatial overlap between the two sets of interactions. We qualified our sentence to “W248 and S338 are located generally outside of the region of the classical E-M coupling”.

4. Pg. 2. Line 32. “Here, we leverage” this sentence is too complicated to understand for readers.

We have revised the sentence to make it easier to understand.

5. Pg. 2. Line 36. Please mention that the novel AO state E-M coupling interactions are from neighboring subunits. Otherwise, readers might get confused.

We agree with the reviewer and have changed it to “When S4 then proceeds on to the fully-activated state, the elbow-like hinge between S4 and S4-S5L engages with the pore of the neighboring subunit to activate conductance.”

6. Fig. 2a. Why so few traces in KCNQ1 W248R currents, when GV shows many points with currents?

Thanks for pointing this out. Due to some endogenous currents in the oocytes, W248R currents above 60 mV were not shown, however, these endogenous currents do not seem to alter the G-V relation. Nevertheless, in the revised manuscript, we deleted the data points above 60 mV in the G-V curve to match the data with current traces. The deletion of the points in G-V curve does not affect the results or conclusion.

7. Fig 2e and 2h. The V50 of G-V is missing or too small in both figures.

We assume the reviewer is pointing Fig 3e,g. The black circles are V50s of G-V, which almost superimpose with that of the F1-V (red circles). We have adjusted the figures to make it clearer.

8. Pg. 10. Line 232. How to get the activation energies of voltage-dependent action of channels should be explained.

We thank the reviewer for pointing this out. The activation energy was given by $\Delta G = -zFV_{50}$, where z is the effective charge, F is the Faraday constant, and V_{50} is the half activation voltage. Both z and V_{50} were obtained by fitting the G-V relation with Boltzmann equation. We have added more detailed explanation in the Methods (see pg. 27).

9. Pg. 14. Line 339 and Pg. 30. Line 604. “K+” should be “K+”.

Corrected.

10. Fig 3c. Missing bar for Kv7.1 in top row.

Corrected.

11. Fig 5e. If possible, color the subunits with another color instead of grey. E.g., it's hard to see the label “S4-S5 linker” in the figure.

As suggested, we have changed the color to make the label clearer.

12. Supplementary Fig3. R243W GV curve shows robust constitutive current. But the current traces don't show any constitutive current. Why is that?

We thank the reviewer for pointing this out. The current traces (including R243W) in sFig. 3a were collected from the regular background, while the data in sFig. 3f (including R243W GV and FV curves) were collected from the VCF background with three more single mutations made for the VCF experiments (pseudo-wild type KV7.1-C214A/G219C/C331A), these mutations have been shown to left-shift the G-V curve and increase the constitutive open current (Osteen et al. 2012 PNAS; Barro-Soria et al. 2013 PNAS; Zaydman et al. 2013 PNAS). We have added sentences to make this clearer in the figure legend.

13. Supplementary Fig.4 line 52. Analyses should be “analysis”

Corrected.

14. Supplementary Fig 5b and e. I would remove the column marked Assigned State. To me that is just confusing.

Done.

15. Supplementary Fig 5e. Shouldn't F232 move relative to F279 when S4 moves up? One would not expect these two residues to always be together in RC, IO, and AO?

We agree with the reviewer that F232 (on S4 segment) and F279 (on S5 segment) should have state-dependent interaction, which was nicely demonstrated by Nakajo and Kubo (Nakajo and Kubo, Nat Commun. 2014). In this study, we calculated the distance between C β atoms of F232-F279, and used a cut-off of 13 Å to determine if these residues are prone to interact when

mutated into Cys. However, this calculation differs from the distance calculation shown in main Fig. 5. The results, 3 4 4 4 over 4 subunits for RC IO AO AC states, suggest the possible state-dependent interactions at F232-F279. We realized that we did not specify the method of our calculation in the legend of Supplementary Fig 5e in the original manuscript, and this is a reason to cause confusion. We now added the method of calculation in sFig 5e legend in the revised manuscript.

16. Supplementary Fig. 6b. Please explain y axis scale? E.g., 3.5 means 10% conserved or 100% conserved? In addition, shouldn't GYG be the most conserved in a K channel alignment?

We apologize for the confusion in the y-axis of SFig. 6b, we will update the legend and methods to appropriately explain this axis.

The y-axis in supplementary is a metric for first order conservation utilizing a statistical measure termed Kullback-Leibler divergence. The details are described in Rivoire et al. 2016. Briefly, K-L divergence quantifies the observed frequency of a sample against a "background" frequency. In pySCA, each amino acid is assigned a "natural" background frequency. SCA calculates the conservation value D_i is then calculated with the equation

$$D_i = \sum_{a=0}^{20} f_i^a \ln\left(\frac{f_i^a}{q^a}\right)$$

Where f_i^a is the observed calculated frequency in the input and q^a is the background frequency for amino acid a . The value D_i is 0 if the observed frequency f is equal to background frequency q , and increases as f deviates from q . D_i does not measure how similar the amino acids are within the alignment, but instead measures deviation of the observed amino acid from expected background. High deviation from background is defined as "positional conservation".

With respect to the signature sequence GYG, the input alignment shows extremely well-conserved GYG (Supp Fig. 6A). Only 42 out of 1421 total sequences (3%) does not feature "GYG" in the alignment. We included these sequences in the alignment based on their annotations in pfam as voltage-gated K⁺ channels, but they represent a very small set of the total sequences.

We have clarified this in the Methods (page 36).

17. Supplementary Fig. 6c. Show the data with arrows better (e.g. Y axis log scale, or inset with finer scale).

We have revised Fig. 6c to show inset with finer scale.

18. Legend Supplementary Fig. 6d. References to Fig 4c, d, and e should be Fig 6c, d, and e.

Thanks for the correction, we have revised the legends.

Reviewer #3 (Remarks to the Author):

In this paper by Hou et al., the authors proposes a two-stage "hand-end-elbow" gating mechanism of Kv7.1 voltage-gated potassium channel. The authors used several experimental and simulation techniques to reach this conclusion. Overall, the manuscript is well written and worthy of publication pending minor to moderate revisions. I have several comments to further improve the manuscript:

Thanks for the positive assessment.

1. The SCA analysis is carried out only on domain-swapped Kv sequences (for good reasons) and is clearly mentioned in the results and the methods sections. However, the abstract reads differently and sounds like this mechanism is applicable to any Kv channel (Page 2, Line 39 and 40). Would not it be better to modify “numerous Kv channels” to “numerous Kv channels with domain-swapped architecture”? We have plenty of evidence now that non-swapped topology channels are gating very differently and soluble/regulatory domains playing very significant role in the process, see for example:

Perissinotti, Laura L., et al. "Determinants of isoform-specific gating kinetics of hERG1 channel: Combined experimental and simulation study." *Frontiers in physiology* 9 (2018): 207

Barros, Francisco, Pedro Domínguez, and Pilar de la Peña. "Relative positioning of Kv11. 1 (hERG) K⁺ channel cytoplasmic domain-located fluorescent tags toward the plasma membrane." *Scientific reports* 8.1 (2018): 15494.

and more recent:

Whicher, J. R., & MacKinnon, R. (2019). Regulation of Eag1 gating by its intracellular domains. *eLife*, 8.

I do feel that emphasizing apparent differences between two topologies present in Kv family may help a lot with application domain of findings reported and will place it into a broader context.

The reviewer makes a very good point, we have revised the introduction/abstract accordingly for appropriate contextualization.

2. Page 17, Line 385: From the text, it seems the cell line measurements are repeated more than 3 times. Can the authors provide further clarification on the reliability of these measurements? Is it reliable and if so, why? It will be helpful for inexperienced reader if this is discussed further in the methods section.

Each electrophysiology experiment was performed on at least 3 individual cells to avoid possible outliers. These experiments are consistently reproducible as shown in the repeated recordings. The number of recordings of each experiment was based on the convention of the field. We have these clarifications in the Reporting Summary, and have now also added into the Methods section.

3. Page 28, Line 548: The reference for CHARMM force field is wrong. The authors mentioned reference is for CHARMM simulation program. Please provide the correct references for CHARMM protein, lipids, ions etc. Please also provide the CHARMM force field versions used.

We appreciate that the reviewer pointed out this important issue. We have corrected the references for CHARMM force field used in this study (Buck et al., 2006; Klauda et al., 2010; MacKerell et al., 1998; Mackerell et al., 2004; MacKerell et al., 2004).

- 1) Buck, M., Bouguet-Bonnet, S., Pastor, R.W., and MacKerell, A.D. (2006). Importance of the CMAP Correction to the CHARMM22 Protein Force Field: Dynamics of Hen Lysozyme. *Biophys. J.* 90, L36–L38.
- 2) Klauda, J.B., Venable, R.M., Freites, J.A., O'Connor, J.W., Tobias, D.J., Mondragon-Ramirez, C., Vorobyov, I., MacKerell, A.D., and Pastor, R.W. (2010). Update of the CHARMM All-Atom Additive Force Field for Lipids: Validation on Six Lipid Types. *J. Phys. Chem. B* 114, 7830–7843.

- 3) MacKerell, A.D., Bashford, D., Bellott, M., Dunbrack, R.L., Evanseck, J.D., Field, M.J., Fischer, S., Gao, J., Guo, H., Ha, S., et al. (1998). All-Atom Empirical Potential for Molecular Modeling and Dynamics Studies of Proteins. *J. Phys. Chem. B* 102, 3586–3616.
- 4) Mackerell, A.D., Feig, M., and Brooks, C.L. (2004). Extending the treatment of backbone energetics in protein force fields: Limitations of gas-phase quantum mechanics in reproducing protein conformational distributions in molecular dynamics simulations. *J. Comput. Chem.* 25, 1400–1415.
- 5) MacKerell, A.D., Feig, M., and Brooks, C.L. (2004). Improved Treatment of the Protein Backbone in Empirical Force Fields. *J. Am. Chem. Soc.* 126, 698–699.

4. Page 28, Line 556: I suppose the authors used SHAKE algorithm. Please mention that.

We used the SHAKE algorithm indeed in order to maintain the bond lengths constant in the water molecules, as well as on the bonds that involve hydrogen atoms. In NAMD program the algorithm used results from the combination of SHAKE and RATTLE algorithms.

5. Page 29, Line 567: Do the authors really mean “spatially constrained”? This might have unnecessary effect on the other parts of the system during production run, in my experience. More reasonable choice would be to use restraints.

We applied harmonic constraints which were gradually released during the equilibration phase of the MD simulation. We maintained these harmonic constraints on the backbone of the selectivity filter during the production phase to prevent ion conduction. We have now changed it to “harmonically constrained”.

Reviewers' Comments:

Reviewer #1:

Remarks to the Author:

I raised three major criticisms to the previous version. In response to my comments, the authors carefully revised the cartoon scheme in Fig.5e (Comment 1), newly added Suppl Fig.6 (Comment 2) as well as Suppl Fig.5g,h (Comment 3). The authors' revisions and responses to these and other points of my comments are all intensive and satisfactory. I have no more specific comments. I highly evaluate the scientific merit and impact of this paper.

Reviewer #2:

Remarks to the Author:

The authors have responded to all of my comments.

Reviewer #3:

Remarks to the Author:

I am very pleased to recommend this excellent paper for publication. I appreciate constructive revisions of the manuscript a lot!

REVIEWERS' COMMENTS:

Reviewer #1 (Remarks to the Author):

I raised three major criticisms to the previous version. In response to my comments, the authors carefully revised the cartoon scheme in Fig.5e (Comment 1), newly added Suppl Fig.6 (Comment 2) as well as Suppl Fig.5g,h (Comment 3). The authors' revisions and responses to these and other points of my comments are all intensive and satisfactory. I have no more specific comments. I highly evaluate the scientific merit and impact of this paper.

Reviewer #2 (Remarks to the Author):

The authors have responded to all of my comments.

Reviewer #3 (Remarks to the Author):

I am very pleased to recommend this excellent paper for publication. I appreciate constructive revisions of the manuscript a lot!